# Transcriptome Profiling of *Oncorhynchus mykiss* Infected with Low or Highly Pathogenic Viral Hemorrhagic Septicemia Virus (VHSV)

**DOI:** 10.3390/microorganisms12010057

**Published:** 2023-12-28

**Authors:** Lorena Biasini, Gianpiero Zamperin, Francesco Pascoli, Miriam Abbadi, Alessandra Buratin, Andrea Marsella, Valentina Panzarin, Anna Toffan

**Affiliations:** Istituto Zooprofilattico Sperimentale delle Venezie, 35020 Legnaro, PD, Italy; fpascoli@izsvenezie.it (F.P.); mabbadi@izsvenezie.it (M.A.); aburatin@izsvenezie.it (A.B.); amarsella@izsvenezie.it (A.M.); vpanzarin@izsvenezie.it (V.P.); atoffan@izsvenezie.it (A.T.)

**Keywords:** NGS, transcriptomic, rainbow trout, VHS, pathogenicity, RNA-Seq

## Abstract

The rainbow trout (*Oncorhynchus mykiss*) is the most important produced species in freshwater within the European Union, usually reared in intensive farming systems. This species is highly susceptible to viral hemorrhagic septicemia (VHS), a severe systemic disease widespread globally throughout the world. Viral hemorrhagic septicemia virus (VHSV) is the etiological agent and, recently, three classes of VHSV virulence (high, moderate, and low) have been proposed based on the mortality rates, which are strictly dependent on the viral strain. The molecular mechanisms that regulate VHSV virulence and the stimulated gene responses in the host during infection are not completely unveiled. While some preliminary transcriptomic studies have been reported in other fish species, to date there are no publications on rainbow trout. Herein, we report the first time-course RNA sequencing analysis on rainbow trout juveniles experimentally infected with high and low VHSV pathogenic Italian strains. Transcriptome analysis was performed on head kidney samples collected at different time points (1, 2, and 5 days post infection). A large set of notable genes were found to be differentially expressed (DEGs) in all the challenged groups (e.s. *trim63a*, *acod1*, *cox-2*, *skia*, *hipk1*, *cx35.4*, *ins*, *mtnr1a*, *tlr3*, *tlr7*, *mda5*, *lgp2*). Moreover, the number of DEGs progressively increased especially during time with a greater amount found in the group infected with the high VHSV virulent strain. The gene ontology (GO) enrichment analysis highlighted that functions related to inflammation were modulated in rainbow trout during the first days of VHSV infection, regardless of the pathogenicity of the strain. While some functions showed slight differences in enrichments between the two infected groups, others appeared more exclusively modulated in the group challenged with the highly pathogenic strain.

## 1. Introduction

Viral hemorrhagic septicemia (VHS) is one of the WOAH-listed diseases responsible for a serious systemic disease in fish. It is caused by a bullet-shaped, enveloped, negative sense single stranded RNA virus belonging to the family *Rhabdoviridae*, genus *Novirhabdovirus* [1].

Viral hemorrhagic septicemia virus (VHSV) genome consists of six genes, ordered 3′-N-P-M-G-NV-L-5′, which encode five structural proteins with conserved functions among Rhabdoviruses [2,3]. The major structural protein is the nucleoprotein (N) that, together with the phosphoprotein (P) and the viral polymerase (L), forms the ribonucleoprotein complex (RNP) responsible for the transcription and replication of the viral genome. Finally, the matrix protein (M) spirals the large RNP complex [4]. The external glycoprotein (G) is the main antigenic protein of VHSV that allows the virus attachment and entry [5,6]. The non-virion protein (NV) is exclusive to the *Novirhabdovirus* genus and completes the protein asset of VHSV. This peculiar non-structural protein has been shown to influence the *in vivo* pathogenicity of the different VHSV [7,8].

Among the several species susceptible to the infection, the rainbow trout (*Oncorhynchus mykiss*) is one of the most severely affected by VHS, with a high level of mortality especially in intensive farming system. The disease is present all over the world, especially in the Northern hemisphere, and in Europe it is endemic in several countries, Italy included [9,10] The phylogenetic studies undertaken so far have described four genotypes (I, II, III, and IV), further divided in several subtypes and clades [11,12,13,14]. In Italy, the disease caused by the VHSV is widespread amongst rainbow trout grow on farms and is responsible for severe economic losses due to fish mortality and commercial constraints [9,15].

It has been demonstrated that virulence in VHSV can mainly depend on the viral genetic makeup [16,17,18]. Indeed, Panzarin et al. [16] have identified an extensive list of polymorphic sites (SAPs), scattered throughout VHSV coding regions, obtained using statistical association analyses that combined *in vivo* virulence records and associated whole genome sequences of known VHSV genotypes. Reported SAPs varied significantly with virulence and, as such, potentially associated with the high and low virulent phenotype. Interestingly, the listed SAPs also included virulence markers located in the N and P protein coding regions, recently suggested to be major virulence determinants for VHSV in rainbow trout [17,19,20]. Moreover, the existence of three different virulence categories of VHSV for rainbow trout, namely low, moderate, and highly virulence, have also been described [16]. Consequently, from this latter study we have selected two VHSV isolates, with distinct virulence classifications, in order to perform the herein presented experiment. Furthermore, *in vitro* studies suggest that virulence can mainly be determined by the capacity of the virus to penetrate the host and spread to target organs, and is independent from replication fitness during the infectious process [21].

Even if the obtained data highlight that VHSV virulence has strong genetic bases, we cannot exclude that the host response to the virus may play a crucial role in determining the outcome of the infection. The aim of the present study is to unravel this aspect, which deserves to be investigated further. In particular, our work wanted to investigate whether rainbow trout transcriptome can vary over time in response to the infection with VHSV strains belonging to two different virulence classes (namely highly and low virulence), as very little information is currently available in the literature on this matter [22,23,24].

## 2. Materials and Methods

### 2.1. VHSV Strains, Isolation and Propagation

Two VHSVs, namely VHSV/O.mykiss/I/TN/80/Mar10 and VHSV/O.mykiss/I/TN/480/Oct96 (GenBank accession numbers MK829682 and MK829677, respectively), stored at −80 °C at the Istituto Zooprofilattico Sperimentale delle Venezie (IZSVe) repository, were propagated in *epithelioma papulosum cyprini* (EPC) cell monolayers for 24 h, according to standard procedures [16]. For each strain, the replication was performed in 75 cm^2^ flasks (CytoOne^®^, STARLAB International GmbH, Hamburg, Germany) at 15 °C and the cell culture supernatant was collected at the appearance of the complete cytopathic effect, then pooled and clarified for 10 min at 2800× g at 4 °C. The RNA was subsequently extracted as described in Section 2.3 and qualitatively checked with TaqMan^®^ reverse transcription real-time PCR (rRT-PCR) for the presence of VHSV as described by Jonstrup et al. [25]; the details of the conditions are summarized in Appendix A. Furthermore, the same samples were also tested for the absence of infectious hematopoietic necrosis virus (IHNV) with methodology and oligonucleotides described by Overturf et al. [26] and infectious pancreatic necrosis virus (IPNV) according to Ørpetveit et al. [27]. Viral titres of cell culture supernatant were calculated and expressed as TCID_50_/mL according to Reed and Müench’s formula [28]. Viral batches were aliquoted and stored at −80 °C until use.

Both the VHS viruses used for the present study were isolated from Italian diseased rainbow trout (*Oncorhynchus mykiss*). Their phenotypes were characterized in a previous work as being highly virulent for VHSV/O.mykiss/I/TN/80/Mar10 and moderate/low virulent for VHSV/O.mykiss/I/TN/480/Oct96 [16]. From now on, we will refer to these viruses as VHSV-H and VHSV-L, respectively, for VHSV/O.mykiss/I/TN/80/Mar10 and VHSV/O.mykiss/I/TN/480/Oct96. These two viruses, together with several others, were also characterized for virulence determinant and intracellular replication by reverse genetics and *in vitro* studies, respectively [21,29].

### 2.2. Mortality and Transcriptomic Experimental Challenges

Female rainbow trout juveniles (*n* = 330) of approximately 13 ± 1 g were purchased from an Italian commercial farm classified within Category I (EU Directive 88/2006), acclimated for a period of 10 days, and tested for main pathogens. During this period, the presence of external parasites was excluded by gills and skin fresh smears observations. Main pathogenic bacteria, such as *Yersinia Rukerii*, *Lactococcus garvie*, and *Aeromonas salmonicidae*, were tested by plating on blood agar according to standard procedures samples from kidney and spleen of ten randomly selected fish. Viruses (infectious pancreatic necrosis, salmonid alphavirus, infectious hematopoietic necrosis, and VHS) were tested by rRT-PCR on internal organs pools (kidney, spleen, and heart) according to already published methods [25,26,27,30]. A commercial feed was supplied for the entire duration of the challenges. Fish were divided in six groups into individual tanks: 3 groups of 55 animals each (for mortality recording) and 3 groups of 25 animals each (for transcriptomic analyses). Groups for mortality and transcriptomic study were paired and challenged by immersion as follows: (i) two groups with VHSV-H, (ii) two groups with VHSV-L, and (iii) two mock-infected groups used for comparison with the infected fish. Exposure with 10^3.05^ TCID_50_/mL of each viral strain was performed for 3 h in 20 L static water, supplying additional aeration. At the end of the challenge, the level of the water was restored to 80 L by adding clean freshwater and the water flow was turned on. For each viral strain, the first group (mortality recording) was monitored regularly for a 4-week period and daily mortality was recorded in order to elaborate survival curves. Spleen (SPL) and head kidney (HK) were collected both from the dead and survivor fish, euthanized by anaesthetic overdose (Tricaine, Pharmaq, Overhalla, Norway) at the end of the observation period. In the second group (transcriptomic challenge), five fish per tank were euthanized by anaesthetic overdose for SPL and HK collection in order to quantify viral genomic copy number (at 0.5, 1, 2, 3, and 5 days post infection (dpi)) and to perform transcriptomic analyses (at 1, 2, and 5 dpi). The third group of both challenges was mock infected with sterile E- Minimum Essential Medium (E-MEM) (Merck KGaA, Darmstadt, Germany) and the same sampling scheme applied to the mortality and transcriptomic groups was adopted.

### 2.3. RNA Extraction, Quality Check and Normalization

Aliquots of the viral batches produced for experimental trials and the organs (SPL and HK pooled for each analyzed fish) of the mortality challenge group were tested for VHSV detection. Total nucleic acids were purified starting directly from 250 μL of viral isolate, while the organs were first subjected to a homogenization step. The specimens were manually grinded in vials using a sterile small potter and quartz sand adding E-MEM to obtain a 1:3 dilution. Homogenized samples were centrifuged at 8000× *g* for 2 min and the clarified supernatants collected. Total nucleic acids were extracted using QIAsymphony DSP Virus/Pathogen Midi kit (Qiagen, Hilden, Germany) in combination with the automated system QIAsymphony SP (Qiagen, Hilden, Germany). Isolation of the nucleic acids was performed following the manufacturer’s recommendations. The RNAs were used undiluted to perform qualitative real time PCR for VHSV diagnosis in order to confirm the cause of mortality, as described below (See Section 2.4).

In the transcriptomic challenge, total RNA was isolated from the spleen and head kidney of each rainbow trout specimen using an RNeasy Mini kit (QIAGEN, Hilden, Germany). The purification protocol was followed according to the manufacturer’s instructions, as described in Toffan et al. [31]. The quantity and quality of all the isolated RNA samples were checked, respectively, using a Qubit^®^ RNA Broad Range (BR) Assay kit (ThermoFisher Scientific, Waltham, MA, USA) and an RNA 6000 Nano kit in Agilent 2100 Bioanalyzer (Agilent Technology, Santa Clara, CA, USA). All the samples with RNA integrity number (RIN) > 9.00 were normalized at 20 ng/µL and subjected to molecular and transcriptomic analysis.

### 2.4. VHSV Detection by Real-Time RT-PCR

In order to confirm the success of infection and determine the cause of death in the mortality recording group, all dead specimens collected during the observation period (*n* = 53 in VHSV-H group; *n* = 2 in VHSV-L group) were tested for the presence of VHSV by a qualitative real-time RT-PCR (rRT-PCR) [25]; for details see Appendix A. Viral positivity was also investigated in survivor fish (*n* = 2 in VHSV-H group; *n* = 10/53 VHSV-L; *n* = 10/55 in mock infected group). All fish samples were analyzed singularly.

### 2.5. In Vitro VHSV RNA Synthesis for Standards Preparation

Reference strain DK-F1 [11] (GenBank accession number MT162452) was used to produce synthetic RNA. The partial nucleocapsid gene was amplified using the Qiagen^®^ OneStep RT-PCR Kit (Qiagen, Hilden, Germany) according to the manufacturer’s instructions, while the primer set used for the RT-PCR is reported in Appendix A. In order to allow subsequent RNA *in vitro* transcription, T7 RNA polymerase promoter sequence was added at the 5′ end of the forward primer. The RT-PCR was performed as described in Appendix A. The PCR product of 470 bp was analyzed for size and purity on 1% agarose gel electrophoresis. Amplicon was then excised from the agarose gel and purified using the QIAquick Gel Extraction Kit (Qiagen, Hilden, Germany), according to the manufacturer’s protocol. *In vitro* transcription reaction was carried out at 37 °C for 4 h using the MEGAshortscript™ T7 Transcription Kit (ThermoFisher Scientific, Waltham, MA, USA) according to the manufacturer’s instructions. The reaction was finished by incubation for 15 min at 37 °C with 1 μL of TURBO DNase. VHSV synthetic RNA was finally purified with the MEGAclear™ Transcription Clean-Up Kit (ThermoFisher Scientific, Waltham, MA, USA) following the manufacturer’s recommendations. The obtained in vitro transcribed RNA (size 453 nt) was therefore quantitated with the Qubit^®^ RNA BR Assay kit (ThermoFisher Scientific, Waltham, MA, USA), supplemented with 40 U of RNAsin Plus RNase Inhibitor (Promega, Woods Hollow, Fitchburg, WI, USA) and preserved at −80 °C until use. The total number of synthetic RNA molecules in the standard stock solution was calculated as previously described [31].

### 2.6. Determining the Limits of Detection and Quantification of the qRT-PCR Protocol

Ten-fold serial dilutions of VHSV *in vitro* transcribed RNA, at known concentrations, were used to determine the limit of detection (LoD) and limit of quantification (LoQ) of the quantitative real-time RT-PCR (qRT-PCR) protocol. The concentration of RNA ranged between 10^−1^ and 10^8^ copy number per µL. Each dilution was tested in triplicate, together with a no-template control. The employed primers and the probe, reported in Jonstrup et al. [25] and targeting a conserved region within the N gene, are described in Appendix A, while the amplification reaction conditions are detailed in Appendix A. The LoD was defined as the lowest sample dilution for which all the replicates tested positive. The LoQ was determined by the lowest sample dilution for which all the replicates tested positive with a coefficient of variation (CV) of the estimated copy numbers < 35 per cent [32].The developed VHSV qRT-PCR protocol provided a LoD and LoQ of 0.5 viral copy numbers (CN) detected in one ng of total RNA.

### 2.7. Quantification of VHSV Genomic Copies

Intra-host viral replication was assessed in fish target organs of the transcriptomic groups. Purified RNAs (*n* = 330) were used to detect the VHSV gene copy numbers by applying a new qRT-PCR protocol. The assay was developed starting from the reference qualitative diagnostic protocol [25] modified with an absolute quantification approach, as described below. Quantitative real-time RT-PCR reactions were carried out in a final volume of 25 μL using 5 μL of template (normalized RNA sample or VHSV synthetic RNA). Amplification reactions were performed in a CFX96 Touch™ Real-Time PCR Detection System (BioRad, Hercules, CA, USA) under the conditions reported in Appendix A. Amplification data were analyzed with the Bio-Rad CFX Manager software V3.1.

The amount of VHSV genome copy numbers was determined by interpolation of the quantification cycle (Cq) values obtained with the standard curve developed with ten-fold serial dilutions of *in vitro* transcribed RNA tested in triplicate. In each run, VHSV synthetic RNA dilutions were used and ranged from 10^2^ and 10^7^ copy numbers per μL. All specimens were analyzed individually (*n* = 150). Each qRT-PCR reaction contained 100 ng of template RNA and the results are reported hereinafter as Log_10_ of copy numbers detected in one ng of total RNA (LCN/ng).

### 2.8. RNA Sequencing (RNA-Seq)

Considering the high costs of the RNA-Seq technique, it was not possible to test all the samples collected from the transcriptomic challenge group. The specific times post-infection (namely 1, 2, 5 dpi), the target organ (namely the HK), and the number of animals (*n* = 4 out of 5) in each of the three groups (challenged with VHSV-H, VHSV-L, and mock-infected) were selected based on the results obtained by qRT-PCR (See Section 3.2 and Appendix A). The transcriptome library of each sample (*n* = 36) was prepared starting from 500 ng of total RNA using the TruSeq Stranded mRNA Library Prep kit (Illumina Inc., San Diego, CA, USA) following the company’s protocols. Each library was run on an Agilent 2100 Bioanalyzer using Agilent High Sensitivity DNA kit to ensure the proper range of cDNA length distribution. Sequencing was performed on an Illumina NextSeq instrument with NextSeq^®^ 500/550 High Output Kit v2 (75 cycles) in single-read mode (Illumina, San Diego, CA, USA) in order to produce about 30 million reads per sample. Raw sequencing data are available at SRA (NCBI) under accession number PRJNA1024374.

### 2.9. Bioinformatic Analysis

Raw data were quality filtered by clipping library adaptors and trimming low quality ends with trimmomatic v0.38 [33] using standard parameters. Passing filter reads shorter than 50 bp were removed. High quality data were aligned against *Oncorhynchus mykiss* genome (Omyk_1.0, Ensembl release GCA_002163495.1) with STAR v2.7.6a [34] using standard parameters. Gene count was generated with htseq-count v0.11.0 [35] using the “union” mode and “exon” feature type. Differential expression analysis was performed with DeSeq2 package v1.20.0 [36]. Differentially expressed genes (DEGs) were defined using FDR < 0.05 and |log2FC| ≥ 1 (FDR: False Discovery Rate; FC: Fold Change). Gene Ontology (GO) terms were assigned to each gene by using Blast2GO v5.2.5 [37] and merged with those downloaded from Ensembl through BioMart [38]. GO enrichment was computed by Fisher’s exact test and *p*-values were adjusted through Benjamini–Hochberg correction. An FDR < 0.05 was considered significant. Enrichment scores were computed as -log(FDR). Child-father relationships belonging to GO graph were reconstructed using the OBO file downloaded from [39]. To compute expression levels of viral genes, we recovered not mapping reads and mapped them to VHSV-H and VHSV-L genomes with STAR v2.7.6a using standard parameters. Gene count was generated with htseq-count v0.11 as previously described. Viral expression levels were computed as RPKM (Reads Per Kilobase of transcript per Million mapped reads.). Expression levels of gene N were used for a comparison with the genomic copy number computed by qRT-PCR protocol. Correlation value was computed as a linear regression value.

### 2.10. Statistical Analysis

To estimate the survival function from lifetime data, the non-parametric Kaplan–Meier method was used, which allowed the survival curve to be drawn as a step curve for each experimental group, measuring the length of time the fish survives the infection [40]. To compare survival curves under different experimental conditions, the Wilcoxon non-parametric test was used. Chi square test was employed to compare cumulative mortality at the end of the study period between independent groups.

The viral quantification data of (five) biological replicates are expressed as mean genomic copy numbers on one nanogram of total RNA. To streamline reading, numbers are reported in the text as logarithmic values of copy number (LCN). For pair comparison of CN from each organ between groups at each time point, Mann–Whitney U test was applied. The level for accepted statistical significance was *p*-value <0.05. Statistical analyses were performed using STATISTICA 13 (Tibco Software, Palo Alto, CA, USA). Reported graphs were generated and edited using GraphPad Prism Software (v10.0.2 GraphPad Software, San Diego, CA, USA).

### 2.11. Ethics Statement

The experimental protocol for fish challenges was designed in compliance with Directive 2010/63/EU and the National Legislative Decree n° 26/2014. The experimental design was evaluated by the IZSVe Animal Welfare Body and Ethics Committee and finally approved by the Italian Ministry of Health with authorization n° 735/2016-PR 22/7/2016.

## 3. Results

### 3.1. Cumulative Mortality and Survivorship Curves

To confirm the difference in terms of virulence of the VHSV isolates in experimentally challenged rainbow trout, cumulative mortality was calculated for each strain over the observation period. Results confirmed the previously established virulence phenotype of the two employed VHSV strains [16], although, in general, a slightly lower mortality was observed. Indeed, at the end of the observation period, cumulative mortality was 96.4% for VHSV-H (*n* = 53 dead; *n* = 2 survivor) and 3.6% for VHSV-L (*n* = 2 dead; *n* = 53 survivor), which confirms the different phenotype of the two VHSVs. All the mock infected fish survived. Kaplan–Meier curves developed by plotting the survival rates during the observation period (Figure 1a) represent the daily mortality data recorded in all the challenged experimental groups. All tested fish resulted positive for VHSV by rRT-PCR.

### 3.2. Determination of Viral Replication

The quantification of VHSV copy numbers was assessed applying the qRT-PCR protocol to the SPL and HK samples from five fish per time point, collected from the VHSV-L and VHSV-H treated groups as well as from the mock group. All detailed values are available in Appendix A. In summary, rainbow trout infected with the VHSV-L strain started to show quantifiable viral loads from 2 dpi (Mean LCN/ng of 1.57 in SPL and 1.61 in HK). Mean LCN/ng values then rapidly started to increase and reached the maximum peak at 3 dpi with values of 2.91 in SPL and 2.70 in HK. At 5 dpi, the mean values slightly dropped to 2.17 and 2.38 in SPL and HK, respectively. On the other hand, quantifiable viral loads in rainbow trout infected with VHSV-H were already detectable starting from 1 dpi, when an increasing trend in viral loads over time was detected. Indeed, at 2 dpi mean LCN/ng values were 4.33 and 4.58 for SPL and HK, respectively, reaching the maximum level of viral genomic copies at 3 dpi (6.23 for SPL; 6.19 for HK). At 5 dpi, the viral loads reached a plateau with observed values of 6.13 in SPL and 6.06 in HK (Figure 1b). As expected, the mock samples analyzed at all time points tested negative for VHSV.

Interestingly, the LCN/ng values detected in VHSV-H infected fish were higher than those observed in fish infected with VHSV-L and exhibited significant differences (*p* < 0.001) at 2, 3, and 5 dpi in both target organs. Moreover, the number of detected viral genomic copies was almost equivalent in the SPL and HK of all the challenged fish. Hence, transcriptome analyses were performed only on HK samples as the head kidney is considered the dominant hematopoietic organ [41,42]. Expression levels for gene N computed with RNA-Seq data showed high level of correlation (R^2^ = 0.9937) with CN values.

### 3.3. Data Quality, Mapping and Normalization

Transcriptome changes in the HK of rainbow trout infected with two different VHSV strains were investigated using RNA-Seq technology. As previously stated, only head kidneys were tested for transcriptome analyses. Sequencing produced a grand total of 1,642,117,765 single-end 75 bp reads for 36 samples, accounting for about 45 million reads per sample (Appendix A). Quality filtering removed 8.4% of the generated data, allowing the recovery of 41,799,829 high-quality reads per sample on average. After alignment of generated high-quality data against *O. mykiss* genome, 79.6% of the reads resulted as uniquely mapping and the intersection with annotation revealed that 22,543,860 high-quality reads per sample on average were assigned to genes.

The principal component analysis (PCA) of normalized counts of host genes confirmed viral infection progression induced by both VHSV strains (Figure 2). In general, samples from infected fish moved along the first PCA component (PC1), which accounts for 46% of the total variance. Transcriptomes from fish infected with a highly pathogenic strain resulted separated from those belonging to fish from the mock infected group, starting from early time points in comparison to what observed for the low pathogenic strain. In detail, transcriptomes from both VHSV-H and -L infected fish at 1 dpi overlapped with those from mocks while, on the second day of infection, a well separation from the mock infected fish was observable only for VHSV-H samples. This trend was further strengthened at 5 dpi. Differently, even if transcriptomes from VSHV-L samples reached the same separation from mocks at 5 dpi, at previous time points (1 and 2 dpi) they showed little to no separation from the mock samples. While looking at the second PCA component (PC2), accounting for 28% of the total variance, no peculiar results were highlighted except for a clear separation of four samples from the remaining ones. More specifically, those samples belonged to various conditions, i.e., one replicate from different time points of mock, low, and high VHSV infected groups. PC2 seemed to capture some aspects of the experiments that are not correlated with the infections.

### 3.4. Transcriptome Analysis over Time

The characterization of host response to the two VHSV strains at the different time points (1, 2, 5 dpi) was performed by comparing infected with mock fish at each time point separately. At 5 dpi, a similar number of differentially expressed genes (DEGs) was detected in both infection challenges, reaching 5338 and 7448 DEGs for VHSV-L and VHSV-H conditions, respectively (Figure 3). However, the latter challenge group also prompted a strong host response at the previous time point, with 2003 DEGs at 2 dpi, and even a mild response at 1 dpi with 866 DEGs. On the contrary, in the VHSV-L challenge group, at 1 and 2 dpi only 106 and 403 DEGs were found, respectively. Such numbers are probably not biologically relevant because they are lower than the variation observed when comparing specific time point mock replicates against each other.

While looking at the gene sharing level within each challenge group, results revealed that 45 and 51% of found DEGs, at 1 dpi, were specific to such time point for VHSV-L and VHSV-H, respectively (Figure 4a,b). When considering 2 dpi, the percentages of specific DEGs were 40 and 27% for VHSV-L and VHSV-H group, respectively. However, at the latest time point (4 dpi), such percentages increased up to 95% for the VHSV-L group, while for the VHSV-H group the percentage reached 76%. Additionally, when considering the VHSV-H specimens, it is worth noting that days 2 and 5 post infection shared 18% of DEGs.

Since pairwise differential expression analysis revealed that during the latest time points the host mounted the largest response, we checked how many DEGs were commonly shared between the two groups (Figure 4c). Overall, the majority of the DEGs were unique to each condition, in particular at 5 dpi, which included 1608 regulated genes (16.5%) for VHSV-L and 3555 DEGs (36.4%) for VHSV-H. About 138 DEGs were shared between all crossed time points of both challenges. Moreover, it is worth mentioning that 2388 (24.5%) DEGs were shared between the two groups at 5 dpi, while 990 (10.1%) DEGs were shared between the three conditions with the highest number of DEGs (2 and 5 dpi VHSV-H, 5 dpi VHSV-L).

For details of all the genes found as DEGs see Appendix A.

### 3.5. Top Differentially Expressed Genes

In order to focus on the genes with a biologically continuous and strong impact in the course of infection, amongst all the DEGs we selected those simultaneously present in all the three sampling time points for each challenge group.

After the removal of the genes without known function, within the VHSV-H group there were globally 39 DEGs, 5 of which were upregulated, 3 downregulated, and 31 displayed a fluctuating expression trend at different time points (see Table 1). The most upregulated gene was *trim63a* (or *MuRF1*) that had a zebrafish orthologous encoding for a RING-type E3 ubiquitin-protein with transferase activity likely involved in positive regulation of DNA-templated transcription. The second over-expressed gene in the VHSV-H group was *acod1* (or *irg1*), predicted to have an aconitate decarboxylase activity resulting in itaconate production. Another upregulated gene in the VHSV-H infected group was *cox-2*, also called *ptgs2*, an enzyme with a peroxidase activity involved in prostaglandin biosynthesis and predominating in inflammation sites.

Moreover, *skia* was found among the upregulated genes. This gene is predicted to have a SMAD binding activity and a transcription corepressor activity.

Lastly, results showed a slight upregulation of a gene that is orthologous to *hipk1* encoding a serine/threonine-protein kinase involved in transcription regulation and TNF-mediated cellular apoptosis.

Considering the downregulation genes, within the VHSV-H group, gene *cx35.4* (or *gjb3*) was found to be the most downregulated. In zebrafish, it encodes a predicted gap junction involved in cell communication. One gap junction consists of a cluster of closely packed pairs of transmembrane channels, the connexons, through which materials of low molecular weight diffuse from one cell to a neighboring cell.

The second downregulated gene was the non-coding gene *mir-142*, a microRNA with a potential regulatory function in eukaryotic cells.

The *zinc finger protein 572* gene, likely involved in the transcriptional regulation as reported for its hypothetical human orthologous, was also slightly under-expressed in the VHSV-H group.

Finally, of the 31 DEGs found as differentially expressed with a fluctuating trend at different time points, we will mention only those which were also found as DEGs in at least two time points of the VHSV-L group. In detail, downregulated genes at 1 dpi but upregulated at subsequent sampling time points were (i) *sox19b*, which encodes a transcription factor DNA-binding protein involved in the development of the central nervous system, while its orthologous *SOX15* evolved through neofunctionalization is required for skeletal muscle regeneration, (ii) *col2a1a*, encoding the alpha-1 chain of type II collagen, and (iii) *tdrd7a*, which encodes a protein with predicted mRNA binding activity. In humans, the Tudor domain-containing TDRD proteins are involved in the post-transcriptional regulation of specific genes required during lens development and also appear to have antiviral activity against herpes simplex virus HSV-1, (iv) *Avd*, encoding for a protein with peculiar biophysical characteristics and highly expressed in the gonads of both male and female zebrafish and in the gills of male fish, (v) *lhx8a*, with an involvement in basal forebrain cholinergic neurons development and in the regulation of transcription by RNA polymerase II, (vi) *kcnk10a*, likely involved in potassium ion transmembrane transport and stabilization of membrane potential, (vii) *tgfa* in zebrafish and human orthologues encodes a ligand involved in epidermal growth factor receptor (EGFR) signaling pathway; it is also involved in positive regulation of cell proliferation, in differentiation, and in development, (viii) *rapgefl1*, in zebrafish and humans, this is a probable guanine nucleotide exchange factor predicted to be involved in G protein-coupled receptors mediated signal transduction and in the development of the nervous system.

*Epyc* (or PG-Lb, dspg3) was also a DEG with a fluctuating trend, but in a different way (downregulated gene at 1 and 2 dpi that changed to upregulated at 5 dpi). This gene encodes a member of the small leucine-rich repeat proteoglycan family involved in fibrillogenesis by interacting with collagenous fibrils and other extracellular matrix proteins.

The only upregulated gene at 1 dpi but downregulated at subsequent sampling time points was *uox*. This gene, in zebrafish and mice, exhibits a urate oxidase activity with an involvement in protein homotetramerization and urate catabolic process.

Within the VHSV-L group, only 10 genes with the same criteria applied to the VHSV-H group were found. One gene was upregulated, two were downregulated, and seven were fluctuant-expressed genes (Table 2). The most upregulated gene was preproinsulin (named *ins*), predicted to have a hormone activity. After post-translational modifications, insulin acts upstream of or within several processes, including regulation of feeding behavior, especially in regulation of carbohydrate and lipid metabolism.

The two downregulated genes were *trim63a*, already described above, and *mtnr1a*. Mammalian orthologue of the latter mentioned gene is melatonin receptor 1A, encoding an integral membrane protein predicted to be involved in G protein-coupled receptor signaling pathway. This receptor likely mediates the major neurobiological effects and circadian actions of melatonin.

Among the differentially fluctuant-expressed genes, at the three time points in the VHSV-L group, the downregulated genes at 1 dpi that were upregulated at subsequent time points were: *col2a1a*, encoding alpha-1 chain of type II collagen, previously described above and *nkx2.5*, predicted to have a DNA-binding transcription factor activity and an essential role during cardiomyogenesis in humans. Moreover, downregulated genes at 1 and 2 dpi but upregulated at 5 dpi were: *klhl41a* likely involved in skeletal muscle development and differentiation and *epyc*, a small leucine-rich repeat proteoglycan. On the contrary, the upregulated gene at 1 dpi that was downregulated at subsequent time points was *uox*, encoding for a urate oxidase protein (see above). Surprisingly, the upregulated genes at 1 dpi and 5 dpi that downregulated at 2 dpi were: a slightly different paralogue of *trim63a*, previously mentioned, and a *dok5*-like gene which in humans encodes for an enzymatically inert adaptor that provides a docking platform for the assembly of multimolecular signalling complexes. Generally, this gene plays a role in activating the MAP kinase pathway and is involved in neuronal differentiation. Moreover, we decided to observe the most up- and down-regulated DEGs of each time point and for each experimental group, regardless of their co-presence in the other time points (Appendix A). Interestingly, we observed that a good amount of those DEGs were already present in the analyses previously described.

### 3.6. Gene Ontology Enrichment Analysis

To better understand the host response to VHSV infection, we examined the enriched biological processes in the differentially expressed gene set using gene ontology (GO) databases (Appendix A). In general, the number of enriched GO terms followed a similar trend to the number of DEGs found at each condition, with both challenges reaching a peak at the last time point (180 and 312 enriched GO terms for VHSV-L and VHSV-H challenge, respectively). The only deviation from such a trend was found at 2 dpi for both challenges. Indeed, at this time point the enriched GO terms accounted for VHSV-L and VHSV-H trials were 135 and 258, respectively, a considerable level of enrichment despite the low number of detected DEGs. It is interesting to note how the low number of detected DEGs (403 in total) at 2 dpi for the VHSV-L challenge boosted a relatively strong enrichment, indicating a probable activation of the host response already at such a time point. At the earliest time point (1 dpi), the VHSV-L group showed no enrichment, while the VHSV-H group displayed only 40 enriched GO terms, thus reinforcing the idea that the host answer mounts later.

Overall, the host answer to VHSV-H infection started early, at 2 dpi, showing an already high number of enriched GO terms (258), which only slightly increased to 312 at the following time point. Considering altogether 2 and 5 dpi, the enrichment scores displayed a clear increasing trend of their values over time, with most general terms represented by “immune system process”, “cell death”, “signaling”, “locomotion”, and “developmental process”. In detail, enrichment scores ranged from 1.3 to 32.2 at 2 dpi and from 1.3 to 55.1 at 5 dpi, averaging, respectively, 5.2 and 5.7. Interestingly, at 5 dpi, specific terms appeared among the other enriched GO terms and included “biological adhesion”, “cytolysis”, “regulation of catalytic activity”, and “lipid metabolic process”.

Surprisingly, also in the VHSV-L challenge, the host seemed to be mounting a response to the infection starting at 2 dpi considering the number of observed DEGs. Most general terms were the same as the ones detected in the VHSV-H challenge, with a notable exception represented by “growth” term, which appeared exclusively enriched at 2 dpi for the VHSV-L group. However, enrichment scores followed a trend similar to the one observed for the VHSV-H challenge, with values that increased from 2 to 5 dpi. In particular, enrichment scores ranged from 1.3 to 12.6 at 2 dpi and from 1.3 to 21.5 at 5 dpi, averaging, respectively, 2.9 and 2.7. At 5 dpi, specific terms appeared among the enriched GO, including “regulation of molecular function”, “immune effector process”, “cell death”, “regulation of catalytic activity”, “intermediate filament-based process”, and “cell adhesion”.

### 3.7. Analysis of Enriched GO Terms as a Graph

Considering the high number of significant GO terms according to the enrichment analysis, we decided to consider the graph structure on the Ontology (Appendix A). Each GO term is a node, and the relationships between the terms are edges between the nodes. In general, a node always has at least one parent term and may have one or more child terms. The relation among GO terms is somehow hierarchical, with ‘child’ terms being more specialized than their ‘parent’ terms, but unlike a strict hierarchy, a term may have more than one parent term. We started from the most specific GO term among the enriched ones and listed all the enriched GO terms that were fathers to it. Such a list of enriched GO terms was considered a “group”. Then we proceeded to select the most specific GO term among the remaining enriched ones, and so on. In this way, we identified 95 groups of enriched GO terms that allowed us to distinguish between shared and virus-specific enriched functions.

Considering the most specific GO terms (Figure 5) resulted enriched at 2 and 5 dpi for both VHSV-L and VHSV-H, many are clearly relative to immune response, e.g., “response to virus”, “regulation of defense response”, “inflammatory response”, “tumor necrosis factor-mediated signaling pathway”, “regulation of apoptotic process”, “cytokine-mediated signaling pathway”, “regulation of cytokine production”, and “regulation of interferon-gamma-mediated signaling pathway”. Other notable GO terms enriched in a similar pattern are “negative regulation of receptor signaling pathway via JAK-STAT”, “regulation of leukocyte migration”, “T cell activation involved in immune response”, “tissue regeneration”, “NAD biosynthetic process”, “regulation of protein kinase A signaling”, “serine family amino acid metabolic process”, and “toll−like receptor signaling pathway”. At 1 dpi, and only for the VHSV-H challenge group, we found enriched “fatty acid biosynthetic process” and “negative regulation of endopeptidase activity”. However, all of them reappeared as enriched at 5 dpi for both challenges.

Conversely, a certain number of GO terms resulted as enriched specifically for only one of the two challenged groups. The enriched biological processes exhibited in the VHSV-H group at both 2 and 5 dpi were “regulation of transcription by RNA polymerase II”, “regulation of histone H3-K4 methylation”, and “protein ubiquitination”, whereas at 2 dpi “antigen processing and presentation of exogenous peptide antigen via MHC class I”, “activation of NF-kappaB-inducing kinase activity”, and “positive regulation of JNK cascade” were enriched. Finally, at 5 dpi, we found ten enriched GO terms: “antigen processing and presentation of peptide antigen via MHC class I”, “peptidyl-tyrosine phosphorylation/dephosphorylation”, “regulation of Wnt signaling pathway”, “glycogen metabolic process”, “angiogenesis”, “regulation of blood pressure”, and three GO involving the “transforming growth factor signaling pathway”.

On the other hand, for VHSV-L challenge, we only found as enriched “negative regulation of NF-kappaB transcription factor activity” at 2 dpi and “CAAX-box protein processing” at 5 dpi.

## 4. Discussion

The purpose of our study was to analyze, at transcriptomic level, the response of rainbow trout infected with high and low virulent VHSV strains. According to the virulence classification proposed by Panzarin et al. [16], VHSVs can be classified as highly virulent when the cumulative percent of mortality (CPM) is >42%, moderate virulent with CPM values ranging from 42 to 14%, and low virulent when CPM is <14%. Indeed, the two VHSVs selected for the present study were characterized as possessing the most diverse phenotype, with the VHSV/O.mykiss/I/TN/80/Mar10 proving to be highly virulent (therefore called VHSV-H) and the VHSV/O.mykiss/I/TN/480/Oct96 low virulent (referred to as VHVS-L) [16,29]. To further confirm the different phenotype of the selected viruses, cumulative mortality experiments were repeated and the obtained CPM, namely 3% for VHSV-L and 96% for the VHSV-H, were as expected.

In our study, we followed the time-course infection of both low and high virulent VHSV by measuring the number of viral genomic copies at different time points (0.5, 1, 2, 3, and 5 dpi) in two important hematopoietic organs, such as the head kidney and spleen [42,43]. No differences in terms of genomic copy numbers between the two target organs at each time point were detected. Such a result allowed to choose a single organ—namely the head kidney as it is considered the dominant hematopoietic organ [43]—for the subsequent transcriptome analyses. Furthermore, genomic copy numbers in both target organs of trout infected with VHSV-H were significantly higher in comparison to the ones found in the group infected with VHSV-L. This could suggest a probable difference in the absorption capacity of the two viruses, as already reported in different *in vitro* studies [21], even if we cannot exclude that other steps in the viral cycle could have been affected by the genomic modifications present in the viral strain used for this study. Indeed, both VHSV infections induced massive changes (10,039 overall annotated genes found to differentially expressed in at least one comparison) in the rainbow trout transcriptome. GO term enrichment analysis showed that they were enriched in 512 GO terms, indicating that VHSV infection modulates the expression of genes involved in a broad variety of functions, mostly related to immune response and metabolism, as expected during a viral infection.

Furthermore, replication experiments highlighted that both VHSV strains increased their viral copies between 1 to 3 dpi and then stabilized between 3 to 5 dpi. However, the VHSV-L viral load increased more slowly, becoming detectable only at 2 dpi, and even less with respect to VHSV-H. Despite the fact that viral replication analysis revealed a similar trend to the infection kinetics, although with different timing among strains, such a trend differed if compared to the transcriptomic analysis. In fact, host response to VSHV-H started at 1 dpi, consistent with viral replication, increased until 3 dpi, and reached its peak at 5 dpi. Differently, VSHV-L stimulated a weaker response until 3 dpi and then increased at 5 dpi, but without reaching the VSHV-H levels regarding the quantity of DEGs, number, and scores of enriched GO terms. Notably, such difference is consistent with the KM survival curve observed for both strains.

Looking at the analyses, we observed that the majority of DEGs always expressed at all time points (*n* = 39) were related to the VHSV-H infection, while only a few (*n* = 10) were specific to the VHSV-L challenge. Worthy of note was the *TRIM63* gene encoding an E3 ubiquitin ligase, which was found to be over-expressed in several vertebrates with a detrimental role during skeletal muscle atrophy [44]. Furthermore, *TRIM63* is also probably involved in other processes (e.g., targeting pro-anabolic factors, intracellular trafficking) and, as other members of its family, seems to have emerging roles in innate immunity, since it is induced by IFNs [45,46]. Interestingly, Kim et al. found *TRIM25*, a protein belonging to the same family as the above mentioned one, among up-regulated DEGs in rainbow trout infected with a highly virulent strain of infectious hematopoietic necrosis virus (IHNV; family *Rhabdoviridae*) [47].

The *acod1* gene was involved in the inhibition of the inflammatory response and acted as a negative regulator of the Toll-like receptors (TLRs)-mediated inflammatory innate response by stimulating the tumor necrosis factor alpha-induced protein TNFAIP3 expression via reactive oxygen species in lipopolysaccharide-tolerized macrophages [48,49]. In zebrafish and humans, *acod1* orthologous genes were involved in the defense response to pathogens and in regulating reactive oxygen species biosynthetic process; however, the implication of these orthologues in promoting viral replication was also demonstrated [49,50]. Interestingly, *acod1* was overexpressed in different hosts, *in vivo* or *in vitro*, following infections with various pathogens (bacteria, virus, or parasite) [51]. In particular, while in some cases an over-expression of *acod1* seemed to promote an antiviral action, in other studies involving vesicular stomatitis virus (VSV, another virus of the family *Rhabdoviridae*) and human alphaherpesvirus 1 (family *Herpesviridae*), *acod1* was associated with the promotion of viral replication [51]. Moreover, it was also observed that the over-expression of *acod1* in organs of mice infected with VSV favored the entry and overgrowth of this viral species by producing itaconate, which promoted protein prenylation via an IFN-I independent mechanism [52]. *Acod1* was also found among the over-expressed genes in a recent study on RNA-Seq conducted on avian influenza virus (Family *Orthomixoviridae*). Interestingly, a transcriptional increase of *acod1* was observed only in the groups infected with precursor strains of highly pathogenic avian influenza viruses [53].

Cyclooxygenase 2 (*cox-2*) had a pivotal role in the establishment and maintenance of inflammatory conditions, as well as in the aetiology of some tumour processes. This gene is also involved in numerous metabolic, signalling, and pathogenic pathways [54]. The over-expression of *cox-2* was considered as an indicator of invasiveness, aggressiveness, and metastatic potential in different malignancies, including carcinomas linked to human papilloma virus (HPV, family *Papillomaviridae*) infection [55,56]. Cox-2 protein was also involved in viral replication in cytomegalovirus (HCMV, family Herpesviridae) infections [57]. In Kaposi’s sarcoma-associated herpesvirus (KSHV, family *Herpesviridae*) infection, *cox-2* was upregulated and involved in the progression of viral infection [58].

*Skia* is another upregulated gene described in our study, probably involved in different processes including transcription, cardiac conduction, cartilage development, and heart looping. Its orthologous in zebrafish played a crucial role in dorsal-ventral patterning embryogenesis [59]. Most importantly, the human orthologous (SKI) was described as a TGF-β signaling negative regulator [60,61] and was found to be differentially expressed in various diseases, as well as *in vitro* studies, which highlighted its upregulation during viral infection [62].

Finally, *hipk1* showed a mild upregulation in the VHSV-H infected group. Interestingly, its human orthologous appeared to interact in p53 regulation during cell apoptosis in acute and chronic kidney diseases [63,64]. As a matter of fact, the kidney is one of the target organs of VHSV infections.

In the VHSV-H group, *cx35.4* turned out to be the most under-expressed gene. Unfortunately, *cx35.4* is not so well characterized; however, other human connexins like genes (*Cx31* and *Cx43*) are widely studied. Interestingly, the downregulation of these latter mentioned genes was observed during a range of human viral infections associated with tumors development [65].

Additionally, the non-coding microRNA-142 was found to be downregulated in the VHSV-H group. Interestingly, its orthologe was observed as downregulated in zebrafish larvae exposed to environmental pollutants [66]. It has recently been established that miR-142 is a specific negative regulator of TIM-1, which is one of the factors involved in the viral internalization in different endothelial disease [67].

During VHVS-L infection the most up-regulated DEG was *ins*, whose role still remains unclear. Conversely, the downregulated genes were (i) *trim63a* belonging to the large family of tripartite motif-containing (TRIM) proteins, known to be crucial for most aspects related to pathogens resistance [45], and (ii) *mtnr1a*, which was actually downregulated only at 2 dpi.

Notably, based on a review regarding the fish innate immunity against *Novirhabdoviruses* [68], the TLR3, TLR7, MDA5, and LGP2 (or *dhx58*) genes were reported as transcriptional up-regulated receptors involved in the recognition pattern in response to VHSV infection. Similarly, we found all the above-mentioned genes over-expressed in fish of both experimental trials, even if at different timings (at 2 dpi for VHSV-H; at 5 dpi for VHSV-L). This observation further strengthened the hypothesis that the infection with the VHSV-L strain induced a slower viral kinetic in comparison to VHSV-H strain.

Looking at the GO terms, in both challenges and as expected during an infection, the host response was mainly represented by the immune system with the activation of biological functions such as “response to virus”, “regulation of defence response”, and “inflammatory response”. Such enriched GO terms appeared while we were performing our experiments during the second part of the study, meaning at 2 and 5 dpi, even though for VSHV-H the enrichment was stronger than the one observed for VSHV-L, as expressed by the enrichment scores (Figure 5). Indeed, functions like “inflammatory response”, “cell chemotaxis”, “response to stress”, and “cytokine-mediated signalling pathway” had almost double the enrichment scores in VHSV-H when compared to VSHV-L infection, and such a result indicated a stronger and earlier immune response induced by the highly virulent strain in comparison to the low virulent one. Notably, some GO terms related to the immune system appeared identical in both strains in terms of timing and score, e.g., “tumour necrosis factor-mediated signalling pathway” and “regulation of interferon-gamma-mediated signalling pathway”. Regarding the latter GO, Kim et al. [47] reported its enrichment at 3 dpi during a similar experimental trial performed using the other salmonid *Novirhadbovirus* (IHNV). Tumour necrosis factor (TNF) is a type II transmembrane protein expressed in the plasma [69]. TNF is a multifunctional cytokine that plays a crucial role in several cellular events, such as cell survival, proliferation, differentiation, and death. This protein exerts its biological functions through the activation of distinct signalling pathways, for example nuclear factor κB (NF-κB) and c-Jun N-terminal kinase (JNK). NF-κB is a major anti-apoptotic cell survival signal, while sustained JNK activation contributes to cell death. The crosstalk between the NF-κB and JNK determines cellular outcomes in response to TNF [63]. The virus ability to modulate TNF-mediated NF-kB activity is generally a strategy to promote viral replication. It has already been described that VHS virus suppresses the early activation of NF-κB in host cells, through the action of NV protein [70].

Immune response is always accompanied by other host processes, directly or indirectly linked to the viral infection. One example could be the need to repair any tissue damage provoked by the viral replication itself or by the immune system activity in an attempt to fight the virus. Indeed, “tissue regeneration” was observed as enriched, with the same score for both strains at 2 and 5 dpi, thus indicating a similar level and timing of tissue repair processes activation. Simultaneously, the GO term “regulation of apoptotic process”, which is a regulated mechanism of cellular death employed by the immune system to counter an infection, together with the “negative regulation of protein serine/threonine kinase activity” appeared enriched for both viruses, although with a higher score during VSHV-H rather than VHSV-L infection. Indeed, serine/threonine kinases are involved in several biological process, including apoptosis. Moreover, other biological functions arose with different timing between the two strains, all of them appearing earlier for VSHV-H compared to VHSV-L. Such functions included some immune system processes, e.g., “toll-like receptor signalling pathway” [71], but were mainly represented by terms such as “negative regulation of endopeptidase activity”, “fatty acid biosynthetic process”, and “NAD biosynthetic process”, all tied to viral needs such as viral entry into the cell, metabolism, and protein modification. In particular, fatty acid metabolism is generally a central process during viral infections. For instance, glycerophospholipid species yields increase during the Influenza virus infection (IVI) in mammalian species [72]. Tanner et al. [73] assumed there was a principal correlation between influenza replication and choline lipids metabolism. As a matter of fact, they found an IVI-mediated reduction in ester-linked phosphatidylcholine (PC) species, as well as an increased level of sphingomyelin (SM) [73] probably connected with expending cellular choline stores for SM synthesis. Such modifications led to an increase in SM species within the infected cell. Higher amounts of SM and short-chain fatty acid-containing ether-linked PC (ePC) species were found in both infected cells and virions and therefore appeared to be involved in viral morphogenesis. On the other hand, it was observed that the levels of long-chain fatty acid-containing ePC had increased in infected cells, while they were lower in the structure of the virion, highlighting the role of this phospholipid in replication of the virus. The increased biosynthesis of fatty acids and membrane lipids may reflect the fact that the virus stores structural lipids for the production of infectious particles. In olive flounder (*Paralichthys olivaceus*), challenged by intraperitoneal infection with VHSV and observed for 7 days (40% cumulative mortality), post infection transcriptome analysis showed that fatty acid degradation was suppressed [74]. This observation led us to speculate that the process of entry and replication of VHSV-H into host cells was completed a few hours after infection, and that the cell machinery was rapidly diverted to favour the pathogen replication. This provoked a high viral genome replication and, as a consequence, impeded the recovery of infected trout despite the strong triggered immune response.

Nevertheless, our analyses revealed the existence of functions that were VHSV-strain specific or displayed an opposite behaviour between high and low virulent strains. In particular, the nuclear factor kappaB (NF-κB) signaling appeared to be regulated differently in the two viral strains: activated for VHSV-H at 2 dpi, as expressed by enriched GO term “activation of NF-kappaB inducing kinase activity”, while inhibited for VHSV-L at the same time point, as expressed by enriched GO term “negative regulation of NF-kappaB transcription factor activity”. Interestingly, the NF-κB is the master transcription factor that controls the cell proliferation, apoptosis, and the expression of interferons, cytokine, and the proinflammatory factors [75]. As NF-κB has a key role in controlling host defense processes, viruses have evolved strategies to modulate NF-κB signaling to either evade host surveillance or exploit it for viral gene expression. In this work, we observed that NF-κB was activated by VHSV-H but negatively regulated by VHSV-L, confirming once again the strongest ability of the VHSV-H strain to infect the host. Moreover, a plethora of functions appeared specifically for VHSV-H in the last time points of infection. However, only some were linked to the immune response, while the majority were related to metabolism, viral entry into the cell, transcription regulation, and cytoskeleton. Such results further confirmed the capacity of VHSV-H in taking control of the cell machinery to promote its own replication. Furthermore, transcription appeared to be regulated at both 2 and 5 dpi, as indicated by GO terms “regulation of transcription by RNA polymerase II” and “regulation of histone H3-K4 methylation”. The latter is an important function to promote transcription, as histone methylation is critical for gene expression through gene accessibility. The transcriptional machinery, RNA polymerase II, is involved not only in the host’s transcriptional cellular processes but also in the viral mRNA capping machinery, as reported for influenza virus [76]. Perhaps a similar involvement of the eukaryotic polymerase could occur during a VHSV infection. Other important biological functions specific to VHSV-H are the de- and phosphorylation of peptidyl-tyrosine, both observed to be associated to cell entry for the herpes simplex virus [77]. Lastly, some GO terms related to blood, such as “angiogenesis”, “vascular endothelial growth factor signaling pathway”, “vascular endothelial growth factor receptor signaling pathway” and “regulation of blood pressure”, emerged as enriched specifically for VHSV-H at 5 dpi. Given that among the major clinical signs of VHSV infection, as indicated by the disease name itself, are hemorrhages caused by the active replication within the vascular endothelial cells [78,79,80], it is not surprising that such GO terms emerged. Similarly, the absence of such hemorrhages, and consequently no mortality, in VHVS-L infected fish is confirmed by the non-presence of these GO terms.

Finally, the only enriched GO term specific to VHSV-L was the “CAAX-box protein processing”. The presence of a CAAX box is known to be a necessary process for the post-translational modification of proteins called prenylation. Briefly, this process involves a lipid addition to nascent proteins that imparts new hydrophobic properties to the mature proteins for targeting towards cell membranes or organellar membranes [81]. It has also been shown that prenylation plays a key role in the life cycle of numerous pathogens, both bacteria and viruses [82,83]. Indeed, the prenylation mechanism of the eukaryotic host cell is exploited to the advantage of the pathogen. This observation could further explain the different virulence of the VHSV strains compared in the present study: the highly virulent virus enters immediately into the host cells, subdues the metabolism machinery to its need and replicates uncontrollably by the immune response. Conversely, the low virulent virus causes infection with a much slower course, encountering more obstacles during cell entry and replication, thus giving the host the opportunity to mount an adequate immune response and resolve the infection in most cases.

## 5. Conclusions

This *in vivo* study investigated the transcriptomic profile modulations in rainbow trout challenged with two VHSV strains belonging to different virulent classes. The abundant amount of DEGs and GO terms confirmed previously published results obtained from *in vitro* experiments or conducted in other fish species. Besides, it has been observed that strains with high or low pathogenicity are responsible for the modulation of many common biological mechanisms but also of some different host responses. Overall, we demonstrated that the processes of inflammatory response, the cytokine-mediated signalling pathway and many genes and processes involved in immune response are altered in rainbow trout during the first days of infection with VHSV. Therefore, we can assert that the achieved results have shed light on the biological and cellular mechanisms activated by VHSV in the early stages of infection; furthermore, they will be of help to future studies whose aim will be to deepen the comprehension and to better clarify the patterns of molecular responses induced by VHSV infection.

## Figures and Tables

**Figure 1 microorganisms-12-00057-f001:**
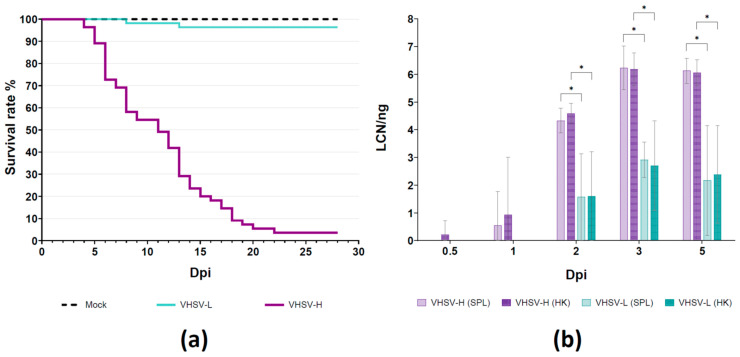
Kaplan–Meier survival curves and qRT-PCR results. (**a**) Mortality challenge: survival curves of VHSV-L e VHSV-H in rainbow trout. Trial carried out by bath immersion. The *y*-axis reports the survival rate; the *x*-axis reports the observation period expressed as days post infection (dpi). Step curves represent the survival rate of challenged fish in each experimental group. (**b**) Transcriptomic challenge: qRT-PCR results. In the *x*-axis the results are grouped according to the sampling day post-infection (dpi). The *y*-axis shows viral target gene copy numbers (CN) detected in SPL or HK and reported on logarithmic scale. Values are expressed as means of normalized CN in 1 ng of total RNA ± SEM (*n* = 5). * = *p*-value < 0.001.

**Figure 2 microorganisms-12-00057-f002:**
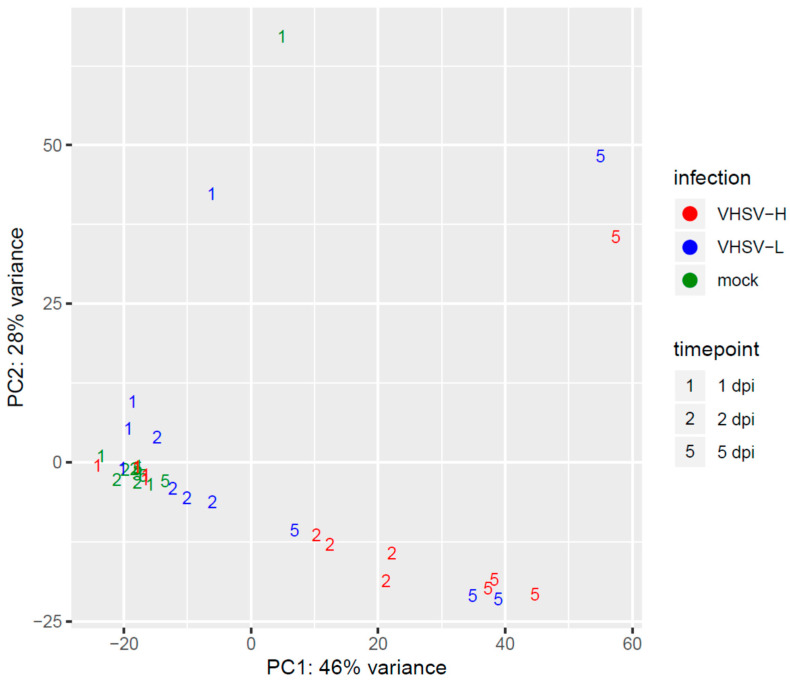
Principal component analysis of the normalized gene counts of the host genes. The *x*- and *y*-axes show the two dimensions that explain the overall amount of variance related to gene expression levels. Each replicate is represented by a colored numerical code based on the challenge group and the time elapsed since infection.

**Figure 3 microorganisms-12-00057-f003:**
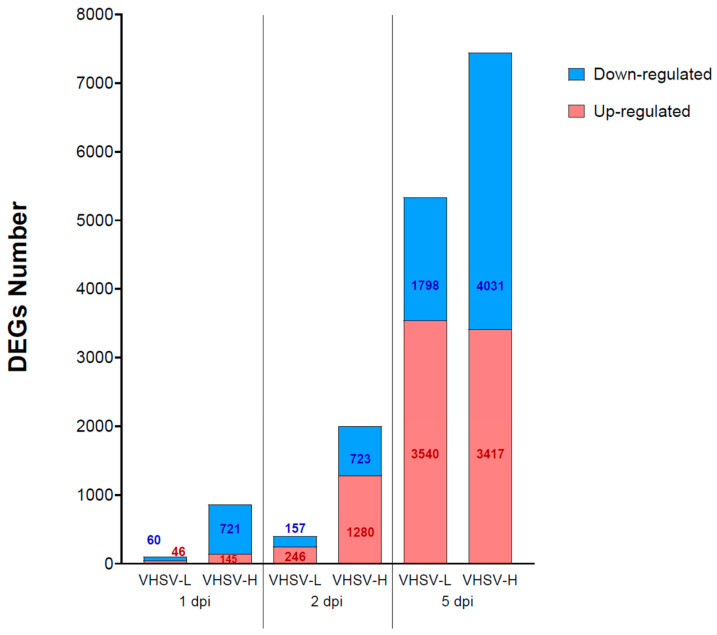
Differentially expressed gene numbers in head kidney of rainbow trout infected with VHSV High or Low virulent strains. At each time point, the total number of genes is shown in a stacked bar chart. Red and blue colors represent up- and down-regulated genes, respectively.

**Figure 4 microorganisms-12-00057-f004:**
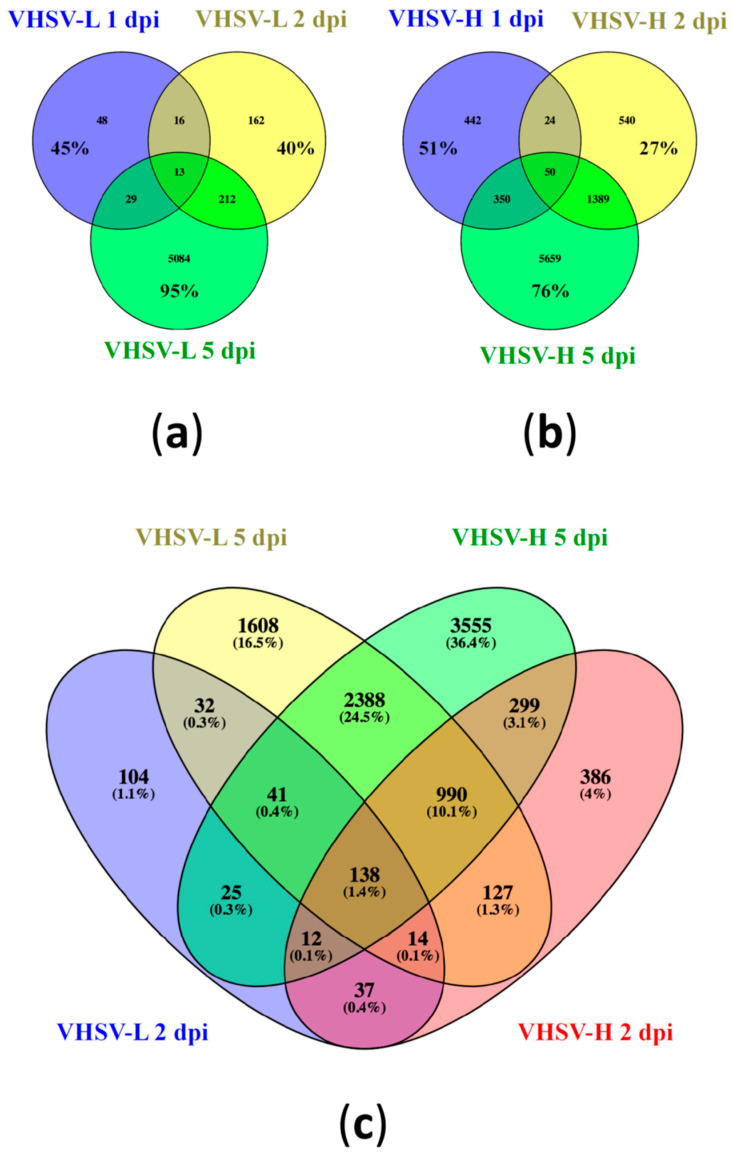
Number of common and specific genes observed after infection. Venn diagrams show: (**a**) the common and specific differentially expressed genes (DEGs) between different time points of VHSV-L challenged group; (**b**) VHSV-H challenged group; (**c**) between the important time points of both challenged groups. Percentages of common and specific genes reported in (**a**,**b**) are computed relative to the total number of DEGs in each time point, while in (**c**) they refer to the total number of DEGs overall the two infections at both time points considered.

**Figure 5 microorganisms-12-00057-f005:**
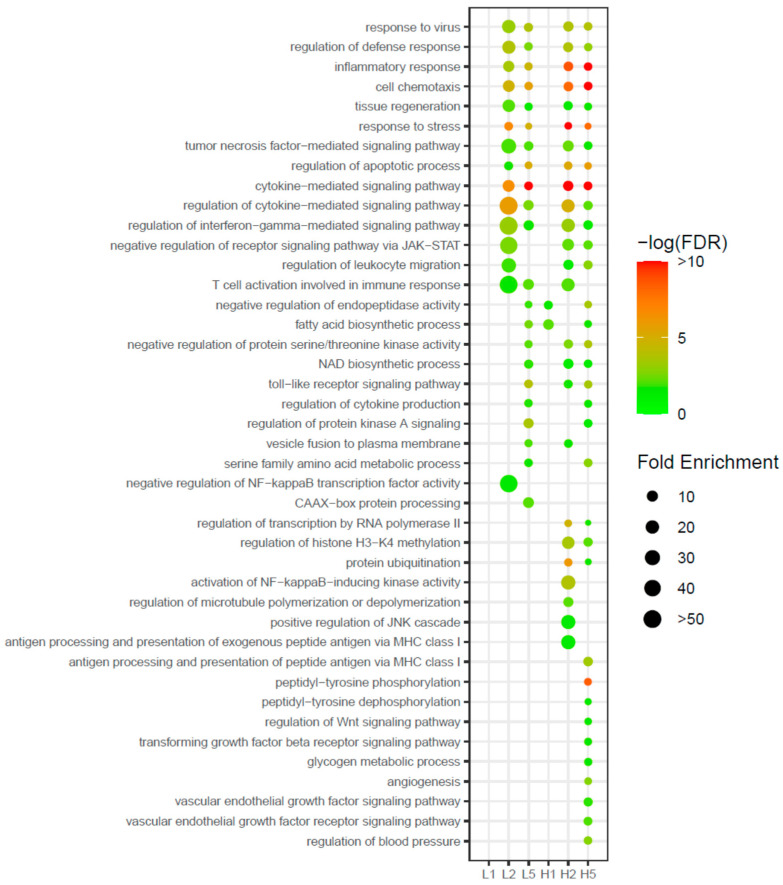
GO enrichment for the L and H VHSV challenges. Dotplot representing the most specific enriched GO terms for all the analyzed time points in both infections. L1, L2, L5 refer to time points 1, 2, 5 dpi in the VHSV-L challenge; similarly, H1, H2, H5 refer to time points 1, 2, 5 dpi in the VHSV-H challenge. Statistically significant enrichments (FDR < 0.05) are presented, and the -Log FDR is shown.

**Table 1 microorganisms-12-00057-t001:** Differentially expressed genes of rainbow trout head kidney in VHSV high virulent strain infected group.

VHSV-H										
Differential Gene Expression Trend	Gene Name	Gene Description	Database Identifier Number					1 dpi log2FC	2 dpi log2FC	5 dpi log2FC
			ZFIN: ZDB-GENE-Id	Uniprot	NCBI: Gene ID	HGNC	miRBase			
Up-regulated	trim63a	tripartite motif containing 63a	040625-139	Q6IQH8 Q969Q1				20.44	18.46	15.09
	acod1	aconitate decarboxylase 1	081104-428	B0UYM1 A6NK06	110501238			3.42	6.50	8.17
	cox-2	Oncorhynchus mykiss cyclooxygenase-2	020530-2	Q9W715 Q8JH43 P35354	100136025			2.62	6.67	3.44
	skia	v-ski avian sarcoma viral oncogene homolog a	990715-9	Q1ECW8 P12755				1.70	1.41	1.88
	Hipk1b	homeodomain interacting protein kinase 1b	140324-1	A0A0R4INQ1 Q86Z02				1.07	1.09	1.45
Down-regulated	cx35.4 (or gjb3)	connexin 35.4 (or junction protein beta 3)	050417-174	Q567J2 O75712				−17.21	−14.68	−14.43
	mir142	microRNA 142				31529	MI0000458	−3.12	−2.84	−5.14
	ZN572-like	zinc finger protein 572-like		Q7Z3I7	110537993			−1.14	−1.34	−1.11
Fluctuating- regulated	hbbe2	Oncorhynchus mykiss embryonic beta-type globin2		Q9YGF4	100135792			−30.69	24.95	28.27
	degs2-like	sphingolipiddelta(4)desaturase/C4-monooxygenase DES2-like		Q6QHC5	110533418			−25.74	−14.96	24.86
	vtg3	vitellogenin-like	991019-2		110521946			−23.89	−24.76	23.14
	sox19b	transcription factor Sox-19b-like	010111-1	Q9DDD7	110500375			−23.85	14.99	22.09
	s100a1	S100 calcium binding protein A1	040916-1	A3KH13	110487342			−22.06	15.92	18.80
	gata4	GATA binding protein 4	980526-476		110522338			−21.72	19.76	18.52
	QRFPR	pyroglutamylated RFamide peptide receptor				15565		−21.06	14.90	19.12
	epyc	epiphycan	041008-9	Q99645				−17.28	−15.66	14.58
	zgc:110045	poly(rC)-binding protein 3-like	050706-149		110537993			−17.23	14.87	−14.61
	col2a1a	collagen, type II, alpha 1a	980526-192	Q2LDA1 B3DLK0				−14.76	14.51	16.17
	tdrd7a	tudor domain containing 7 a	040426-2103	A6NAF9 Q8NHU6	406379			−10.82	21.93	24.81
	avd	avidin	140205-1	A0A8M1P1N5				−10.44	18.22	9.14
	si:dkeyp-46h3.6	L-rhamnose- binding lectin CSL3-like	060503-90	A3KQ75	110528627			−9.61	17.00	9.69
	lhx8a	LIM/homeobox protein Lhx8-like	031008-2	Q6BDC3	110508831			−8.80	16.19	7.41
	kcnk10a	potassium channel subfamily K member 10-like	041210-291	A3QJX1	110522533			−7.26	16.80	6.92
	tgfa	protransforming growth factor alpha-like	040724-208	Q7T011 P01135	110494101			−6.60	13.76	5.39
	lama3	laminin subunit alpha-3-like	050208-475	A0A8M9Q7C5	110521524			−6.11	4.81	4.63
	anxa13l	annexin A13-like	050522-310		110509080			−4.18	3.01	3.30
	rasl11b	ras-like protein family member 11B	040426-793	Q6P0U3	110521663			−3.46	2.82	4.58
	stap2b	signal-transducing adaptor protein 2-like	040426-2540	Q6NYF8	110524344			−3.09	2.44	3.47
	chga	chromogranin A			110529746			−2.77	2.38	1.65
	ptpn3	protein tyrosine phosphatase non-receptor type 3	030131-2934	F1QLN8				−2.53	3.02	3.84
	RND3	rho-related GTP-binding protein RhoE		P61587	110501874	671		−2.36	2.56	2.96
	rapgefl1	rap guanine nucleotide exchange factor-like 1	131121-404	E7FGN2 Q9UHV5	110486183			−2.31	3.97	6.18
	exoc3l4	exocyst complex component 3-like protein 4		Q17RC7	110497811			−2.11	4.32	4.61
	tmem106b	transmembrane protein 106B-like		Q9NUM4	110486388			−1.65	1.56	2.98
Fluctuating- regulated	spsb4a	SPRY domain-containing SOCS box protein 4-like	070911-3	B0S5W3	110508890			−1.34	1.71	2.70
	mycbp	C-Myc-binding protein-like			110496688			−1.29	1.86	2.41
	rnf217	probable E3 ubiquitin-protein ligase RNF217	060503-400	Q1LUI6	110529563			1.43	−1.46	−2.80
	tgm2l	protein-glutamine gamma-glutamyltransferase2-like	050420-97	A3KQ87	110520123			17.71	−19.21	20.16
	uox	uricase-like	030826-24	Q6DG85	110521955			26.18	−27.02	−26.79

**Table 2 microorganisms-12-00057-t002:** Differentially expressed genes of rainbow trout head kidney in VHSV low virulent strain infected group.

VHSV-L								
Differential Gene Expression Trend	Gene Name	Gene Description	Database Identifier Number			1 dpi log2FC	2 dpi log2FC	5 dpi log2FC
			ZFIN: ZDB-GENE-Id	Uniprot	NCBI: Gene ID			
Up-regulated	ins	preproinsulin	980526-110	O73727 P01308	30262	20.38	16.27	16.99
Down-regulated	mtnr1aa	melatonin receptor Mel1A a	990415-155	P51046 P48039	100135888	−16.57	−16.20	−15.24
	trim63a	Oncorhynchus mykiss muscle RING finger 1	040625-139	Q6IQH8 Q969Q1	100505417	−15.47	−17.53	−14.95
Fluctuating- regulated	uox	uricase-like	030826-24	Q6DG85	110521955	26.71	−27.11	−27.43
	trim63a	tripartite motif containing 63a	040625-139	Q6IQH8 Q969Q1		17.84	−18.35	16.48
	dok5	docking protein 5-like		Q9P104	110528565	14.46	−14.79	14.63
	klhl41a	kelch-like family member 41a	081105-22	E9QIN8 O60662	100148315	−16.59	−15.36	14.91
	epyc	epiphycan	041008-9	Q99645		−16.99	−15.75	16.37
	col2a1a	collagen, type II, alpha 1a	980526-192	Q2LDA1 B3DLK0		−14.67	14.56	17.42
	nkx2.5	homeobox protein Nkx-2.5-like	980526-321	Q98872 P52952	110489113	−14.71	15.04	15.91

## Data Availability

RNA sequencing raw data generated in the present study are available at SRA (NCBI) under accession number PRJNA1024374.

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
