# Peer review of "Transcriptome Profiling of Oncorhynchus mykiss Infected with Low or Highly Pathogenic Viral Hemorrhagic Septicemia Virus (VHSV)"

_microorganisms, 2023, doi:10.3390/microorganisms12010057_

Round 1
Reviewer 1 Report
Comments and Suggestions for Authors
The manuscript is focused on the transcriptome analyses pf rainbow trout infected by VHSV. A large set of notable genes 23 were found as differentially expressed (DEGs) in all the challenged groups (trim63a, acod1, cox- 24 2, skia, hipk1, cx35.4, ins, mtnr1a, tlr3, tlr7, mda5, lgp2). The gene ontology (GO) enrichment analysis highlighted that functions 27 related to inflammation were modulated in rainbow trout during the first days of VHSV infection. Those data are clear and enough to publish to Microorganisms journal.
This paper is a transcriptome analysis study of rainbow trout for VHSV infection, and the experimental method and data analysis were neatly organized and analyzed. Key gene expression patterns were derived through DEG analysis, and it is judged to be reasonable to infer the biological functions of genes using GO analysis using those genes as the core. It would be the icing on the cake if the genetic mechanism for viral infection could be derived beyond the inference stage, but such research is expected to be conducted in the next study. Therefore, in this paper, I judged that transcript profiling to infer gene expression patterns and functions for viral infection would be sufficient.
Reviewer 2 Report
Comments and Suggestions for Authors
Manuscript Number: microorganisms-2770515-peer-review-v1
Title: Transcriptome profiling of Oncorhynchus mykiss infected with Low or Highly Pathogenic viral hemorrhagic septicemia virus (VHSV)
Review:
In this manuscript Biasiniet al., analyze by RNA seq the response of rainbow trout to VHSV at different time points after infection using low and high virulent viral strains. This manuscript is of great interest since this experimental design allows the authors to understand how the viral infection proceeds and potentially discover the mechanisms involved in the immune response against VHSV.
Authors have done a great effort in the analysis of the absolute quantification of the viral load in the fish tissues. The text is well written and the objectives are clear. However, the analysis of the transcriptomic data should be improved. The analysis of DE genes should be developed more in detail by using other alternative criteria to select DE genes. I suggest to describe first the GO terms (a general idea of the processes) and then describe the DEGs. Also, the RNAseq results should be validated by qPCR.
Following I have some specific comments that should be considered.
Introduction:
This section is well explained and the objectives are clearly indicated. I only consider that the authors should explain more in detail the information about the virulence (lines 58-60). It is not clear what is the most important genetic characteristic that drastically affects the viral pathogenesis.
I am not sure if the classification of low, moderate and high pathogenic strains used in the ref 16 is also used to describe the Italian stains used for the infection (lines 67-69). It will be nice for the reader to have this information just in the introduction section (although can also appear in the methodology)
Materials:
Line 73: Please, separate “Two VHSVs”
Lines 80-83: The methodology to detect the presence of VHSV and the absence of IHNV and IPNV should be briefly described. Real time PCR, real time reverse transcription and viral detection by using Taqman probes have been used in this section. Please, include additional information.
Line 95: Fish were tested for main pathogens. Please, specify which pathogens were tested and how.
Lines 108-109: Authors comment that “5 fish were euthanized by anesthetic overdose for SPL and HK collection in order to perform molecular analyses (at 0.5; 1; 2; 3 and 5 days post infection (dpi). Please, replace molecular analysis by determination of viral load”. The term “molecular analysis” makes some confusion to the reader.
Line 116: Please, specify that head kidney and spleen were used
RNAseq validation: Although there is not a clear consensus about the RNAseq validation, nowadays almost all papers using RNAseq data includes the validation of the results by using qPCR. I suggest to select a few DEGs and check the expression levels by qPCR.
Results:
Figure 1 and 2 should be combined in a multi-panel figure 1. It will be easy for the reader to see this information in the same figure. It will be easy to observe that the increment of viral load 5 days after infection matches with the beginning of the mortalities in fish. It makes more sense to have all this information in a single figure.
Line 291: “As previously stated, spleens were not tested for transcriptome analyses”. In my opinion authors should delete this information in the manuscript. You should delete from the material section to match exactly with the results presented. If you do not analyze this sample (spleen), please, delete form the methods and results. This will simplify the methods and make clear the reading.
Line 299: PCA analysis. What could be reason of samples that are not grouped (outliers)? Did you check the sex of the animals?
Line 321: Transcriptome analysis over time. It would be interesting to check the increment of viral load using RNAseq data (although the authors have already prove the viral replication in the HK)
Line 360:
In my opinion the analysis of GO terms should be before the analysis of specific genes. Authors should describe first a general idea of the processes that are modulated and following to explain the implication of specific genes
Lines 332-348: The PCA analysis reveal that the expression of genes taken at day 1 and 2 are similar (dots are closer) and the differences appear at day 5, when the viral load is higher (according to the figure 2). Moreover the PCA analysis also revealed that genes from VHSV-L are grouped at 1 and 2 dpi while VHSV-H at day 2 are clearly separated from day 1. It could be also possible due to the viral load. In this context I suggest the authors to analyze genes that are highly modulated although they were not present in all the sampling points. Looking at the Venn diagrams in figure 5 it is clear that there are high number of genes (and processes) that are time dependent and appear only at specific time points. This is why I suggest the authors to include new analysis of the data.
Lines 362-364: I understand that there is high number of comparisons that provide interesting information about the immune response induced by the virus. Authors selected “those DEGs simultaneously present in all the three sampling time points for each challenge group”. This is right but could exclude relevant physiological processes (including immune pathways) that depend on the viral load. As I comment above, I suggest including additional analysis of RNAseq data.
Lines 365-444: Please remove references from this section. Authors must describe the results. The description of the genes and their functions should be included in the discussion section. As I comment before, in my opinion it is easy for the reader to know first which biological processes are modulated (activated/repressed) by the infection (a general idea) and then explain the genes involved in those processes.
In my opinion this part of the manuscript should be revised in detail. All this information is already available in the table 1. It is surprising to me that the authors only focused on genes that appear in all sampling points and after filtering the data they only describe 39 genes for VHSV-H and 10 genes for VHSV-L. At this point I have no clear idea about which physiological processes are involved in the response to VHSV regardless of their pathogenicity.
Lines 451: It is not clear which DEGs were used in this analysis. Did you use all the DEGs obtained in each sampling point? or did you use the selected genes described in the previous section (39 genes for VHSV-H and 10 genes for VHSV-L). Reading this section I assume that you use all the DEGs obtained for H and L. In my opinion those analysis are difficult to combine with the previous description of selected genes since different set of genes were used.
I suggest explaining the GO terms before the description of particular genes.
Lines 451-525: This section is too long. Authors should mix the numerical information with the results presented in the graph and delete all the duplicated information. The discussion of the results must be moved to the next section. In my opinion it is obvious that the increment of DEGs (figure 4) will also produce an increment of the GO terms in the enrichment analysis. I think that section 3.6 should be deeply reduced and combined with the next section (3.7)
Lines 489-495: This part should be deleted or reduced.
This is the final part of the results. At this point authors described the infection process (mortalities and viral load), the biological processes that are activated after infection (enrichment analysis) and the genes that are present in all sampling points. I think that the information about specific genes is not well integrated in the text. Authors should analyze the set of genes with other comparison to have a better understanding of the infection process.
I wonder which biological processes are active during VHSV-H and L infection? Are they the same? Do they appear at the same time point or at different times? Could they be activated when the viral load reaches a minimum amount? Some of these questions are described in the text.
Which genes are involved in those processes? This information is not explored in the text since authors only analyze genes presented in the 3 sampling points. The description of genes should be improved by using additional comparisons. This information could be included in additional figures.
In my opinion figures 3, 4 and 5 should be combined in a unique multi-panel figure.
Discussion:
Line 546: Too many references to describe the HK and spleen as the main hematopoietic organs. Please, review the references and select the most important.
Lines 546-549: This result is quite interesting since it reflects that both organs are targets for the viral infection. It is frequently said that the HK is the main target organ, but your results also suggest that VHSV replicates in the SPL as efficient as in the HK.
Lines 551-552: Differences in viral load between H and L could be related with different ability to replicate inside the fish. Why did the authors specify that the “absorption capacity” is altered? It could be possible that other steps in the viral cycle be affected by the genomic modifications described in those viral stains? Did the in vitro studies in reference 19 use the same viral strains described in the present manuscript?
Line 554: “VHSV-L and VHSV-H strains most likely activates the host response in a different way”. This sentence is already supported by the data presented in the manuscript (figure 6). VHSV-H induce several GO processes that are not activated by VHSV-L.
Line 559: Please delete “cellular”
Line 560: Why did you specify “systemic infection”? Did you check the presence of VHSV in almost all fish organs? I suggest delete the “systemic” term.
Line 573: “VSHV-H starts to shape the host cell metabolism much earlier by requesting supplies for its own replication and to provoke the fish death”. Is this statement supported by the data? Which kind of metabolism is altered by the viral infection? Please, specify it.
Line 575:I think that this is the weakness of the manuscript. Authors only analyze the DE genes that were expressed in all time points. Authors could analyze all the exclusive genes expressed at the different time points (in H and L infection) and see which GO terms are enriched in. Moreover they could compare the processes in H and L, which are specific for H and do not appear in L…Some of this information is already included in the manuscript.
Lines 557-580: is any previous study to support the participation of the TRIM63 gene in the anti-viral response?. I think that information in the lines 578-580 is not of great interest (heart) but TRIM63 and TRIM family proteins have been described to have emerging roles in innate immunity, since they are induced by IFNs (Ozato et al., 2008: doi: 10.1038/nri2413; Li et al., 2019: doi: 10.1016/j.prp.2019.152573).
Overall, the discussions section is well developed and it is easy to follow. I only want to highlight that almost all the description is speculative and is based on the bibliographic information using other viral and animal models. The discussion of the present manuscript could be stronger if authors include experimental data that support most of the suggestion.
Author Response
In this manuscript Biasini et al., analyze by RNA seq the response of rainbow trout to VHSV at different time points after infection using low and high virulent viral strains. This manuscript is of great interest since this experimental design allows the authors to understand how the viral infection proceeds and potentially discover the mechanisms involved in the immune response against VHSV.
1) Authors have done a great effort in the analysis of the absolute quantification of the viral load in the fish tissues. The text is well written and the objectives are clear. However, the analysis of the transcriptomic data should be improved. The analysis of DE genes should be developed more in detail by using other alternative criteria to select DE genes. I suggest to describe first the GO terms (a general idea of the processes) and then describe the DEGs.
Authors’ response: We thank the reviewer for the suggestion. However, we prefer to maintain the current order since we strongly feel that it is the most useful and consistent order to present the results. The main reason for this choice is related to the fact that the order of the performed analysis is: 1) firstly differential expression analysis, to find out which genes change their status by being activated (up-regulated) or switched off (down-regulated); then 2) enrichment analysis, to find out which biological processes (represented by gene ontology terms) are significantly interested by genes differentially expressed found out previously. Therefore, DE genes naturally come before GO terms and, consequently, it is more consistent the current order respect the ones which present GO terms firstly. Another strong reason to maintain the current order is due to the type of research we conducted, namely transcriptomics. This is ones of the many -omics disciplines (e.g. metabolomics, genomics, proteomics, Methylomics, etc.) that study huge groups of factors together as a whole and try to sum up the net results of all such factors acting simultaneously. In our case, by choosing transcriptomics, we studied the expression of about 40,000 genes of rainbow trout at the same time with the purpose of figuring out a neat result and not focusing on the behaviour of few genes. In this view, we firstly described DE genes in general and highlighted the behaviour of some of them selected with a criteria that was the most fitting to our aims (i.e. to find differences between high and low virulent VHS viruses), secondly we sum up the neat result of all genes (DE and not DE) by using GO terms associated to rainbow trout genes and performing enrichment analysis on them using DE (and not DE) genes.
2) Also, the RNAseq results should be validated by qPCR.
Authors’ response: We agree with the reviewer that a validation by qPCR on a set of selected genes could be a good addition. However, nowadays gene expression levels estimated by RNA-Seq are considered reliable and not require to be further validated by other methods (https://www.ncbi.nlm.nih.gov/pmc/articles/PMC5436640/). In fact, it has been more than 10 years that RNA-Seq exists and it is true that, at the beginning, validation using an independent technique, (e.g. qPCR, which is recognisable and statistically assessable (https://pubmed.ncbi.nlm.nih.gov/21498551/)) was required. However, qPCR validation of RNA-Seq data has generally shown that different RNA-Seq methods are highly correlated (https://pubmed.ncbi.nlm.nih.gov/18451266/, https://pubmed.ncbi.nlm.nih.gov/19056941/, https://pubmed.ncbi.nlm.nih.gov/20368969/). Moreover, developing a qPCR assay for multiple genes on an organism like Oncorhynchus mykiss, which lacks available resources of quality comparable to known organisms like human, mouse and rat, it is not an easy task and require many efforts on his own that are more than enough for a separate work. For these reasons, we choose to not include any qPCR validation on our work and leave them to future studies that could focus on precise aspects, e.g. on particular genes or biological processes of interest highlighted by the current work that was a first effort in investigating an unexplored field.
Following I have some specific comments that should be considered.
Introduction:
This section is well explained and the objectives are clearly indicated. I only consider that the authors should explain more in detail the information about the virulence (lines 58-60). It is not clear what is the most important genetic characteristic that drastically affects the viral pathogenesis.
The text was edited according to the Reviewer’s comment (lines 59-66).
I am not sure if the classification of low, moderate and high pathogenic strains used in the ref 16 is also used to describe the Italian stains used for the infection (lines 67-69). It will be nice for the reader to have this information just in the introduction section (although can also appear in the methodology)
The text was edited according to the Reviewer’s comment (lines 66-70).
Materials:
Line 73: Please, separate “Two VHSVs”
The text was edited.
Lines 80-83: The methodology to detect the presence of VHSV and the absence of IHNV and IPNV should be briefly described. Real time PCR, real time reverse transcription and viral detection by using Taqman probes have been used in this section. Please, include additional information.
Authors’ response: the methods used are described in details in the cited paper (namely Jonstrup et al 2013 for VHSv; Ørpetveit et al 2010 for IPNv and Overturf et al 2001 for IHNv). For readability reasons and for keeping M&M concise, we do prefer not detail anymore the methods used. However, we have added some more explicit information in the text (Line 89-95)
Line 95: Fish were tested for main pathogens. Please, specify which pathogens were tested and how. Authors’ response: The pathogens tested as well as the diagnostic methods used have been included in the manuscript (lines 112-119).
Lines 108-109: Authors comment that “5 fish were euthanized by anesthetic overdose for SPL and HK collection in order to perform molecular analyses (at 0.5; 1; 2; 3 and 5 days post infection (dpi). Please, replace molecular analysis by determination of viral load”. The term “molecular analysis” makes some confusion to the reader.
The text was edited according to the Reviewer’s comment (lines 133-134).
Line 116: Please, specify that head kidney and spleen were used
The text was edited according to the Reviewer’s comment (lines 139-142).
RNAseq validation: Although there is not a clear consensus about the RNAseq validation, nowadays almost all papers using RNAseq data includes the validation of the results by using qPCR. I suggest to select a few DEGs and check the expression levels by qPCR.
Authors’ response: We thank the reviewer for the suggestion. As explained at the beginning (comment number 2 – “Review” section) we chose to not perform any RNA-Seq validation on a set of selection genes due to multiple reasons which can be summarized as follow: high efforts required and not really necessary. In fact there are a lot of published papers using RNA-Seq data which do not included any validation at all; even works related to the recent Covid-19 pandemics employed RNA-Seq without any qPCR validation (https://www.nature.com/articles/s41586-022-05282-z). For this reason, we choose to rely only on RNA-Seq data which, over the years, were proven to be reliable.
Results:
Figure 1 and 2 should be combined in a multi-panel figure 1. It will be easy for the reader to see this information in the same figure. It will be easy to observe that the increment of viral load 5 days after infection matches with the beginning of the mortalities in fish. It makes more sense to have all this information in a single figure.
The figures were modified according to the Reviewer’s comment.
Line 291: “As previously stated, spleens were not tested for transcriptome analyses”. In my opinion authors should delete this information in the manuscript. You should delete from the material section to match exactly with the results presented. If you do not analyze this sample (spleen), please, delete form the methods and results. This will simplify the methods and make clear the reading.
Authors’ response: The Authors wish to thank the Reviewer for the comment. However, we retain appropriate and interesting for readers to maintain all results regarding the spleen (meaning only RT-qPCR results), as the initial part of our study focused on investigating which was the viral target organ among the head kidney (HK) and spleen (SPL) through performing quantitative real time PCRs. Indeed, HK and SPL are known to be two important hematopoietic organs in rainbow trout. Once we have observed an almost equivalent viral replication trend in both organs of all the challenged fish, hence we have decided to perform RNAseq, and further transcriptome analyses, using only the HK samples as it is considered the dominant hematopoietic organ (Zapata et al., 2006; Bjørgen et al., 2021). Moreover, our choice on using only HK was also based on the availability of another published manuscript regarding transcriptome analyses of olive flounder head kidney infected with two VHSV strains (moderate and highly virulent) (Hwang et al., 2018).
Please take in to account these our explanations also in response to your interesting comment down reported “Lines 546-549: This result is quite interesting since it reflects that both organs are targets for the viral infection. It is frequently said that the HK is the main target organ, but your results also suggest that VHSV replicates in the SPL as efficient as in the HK.”
However, for more clarity we have changed the sentence “As previously stated, spleens were not tested for transcriptome analyses” to "As previously stated, spleens only head kidneys were not tested for transcriptome analyses" (Line 347).
Line 299: PCA analysis. What could be reason of samples that are not grouped (outliers)? Did you check the sex of the animals?
Authors’ response: sex of the animals has been checked. The farm which provided the fish produces all female juveniles and thus all fish used for the trial were females. This feature, that could influence the work outputs, is not relevant since it’s representative of the field. In any case, sex of the animals is not responsible for outliers.
Line 321: Transcriptome analysis over time. It would be interesting to check the increment of viral load using RNAseq data (although the authors have already prove the viral replication in the HK)
Authors’ response: We thank the reviewer for the suggestion. We computed expression values (RPKM) for the six genes encoded by VHSV (G, L, M, N, NV, P) and compared them to those of the qPCR used to check the viral replication. Since there was complete agreement between the two series of values (R2=0.9937) we decided to not add this information to the paper but, since you noticed it could be interesting, we added values for N gene to supplementary table S3, description on material and methods and a sentence to paragraph 3.2 (Lines 326-327).
Line 360: In my opinion the analysis of GO terms should be before the analysis of specific genes. Authors should describe first a general idea of the processes that are modulated and following to explain the implication of specific genes
See response to the comment number 1 above in the “Review” section.
Lines 332-348: The PCA analysis reveal that the expression of genes taken at day 1 and 2 are similar (dots are closer) and the differences appear at day 5, when the viral load is higher (according to the figure 2). Moreover the PCA analysis also revealed that genes from VHSV-L are grouped at 1 and 2 dpi while VHSV-H at day 2 are clearly separated from day 1. It could be also possible due to the viral load. In this context I suggest the authors to analyze genes that are highly modulated although they were not present in all the sampling points. Looking at the Venn diagrams in figure 5 it is clear that there are high number of genes (and processes) that are time dependent and appear only at specific time points. This is why I suggest the authors to include new analysis of the data.
Lines 362-364: I understand that there is high number of comparisons that provide interesting information about the immune response induced by the virus. Authors selected “those DEGs simultaneously present in all the three sampling time points for each challenge group”. This is right but could exclude relevant physiological processes (including immune pathways) that depend on the viral load. As I comment above, I suggest including additional analysis of RNAseq data.
Authors’ response. The Authors wish to thank the Reviewer for both the comments which we will answer together. The PCA is a sort of preliminary analysis performed in order to check how samples are separated between different conditions and how biological replicates cluster together within the same condition. The read counts of all the genes are combined together to create a set of coordinates on multiple dimensions for every sample. For visual purposes, only the coordinates of the first two dimensions (i.e. those who explain the most variance of the dataset) are used in the graph, so the Figure 2 is a very preliminary representation of the dataset, way before computing differentially expressed genes or enriched gene ontology terms. Therefore, basically, it is not possible to analyse genes highly modulated in the context of the PCA. Apart such technical reasons, very few information are available regarding VHSV infection and our work on this topic has to be considered a pioneering one, so our main aims were to provide a view on the overall progression of the infection from the host view and to focus on the general different behaviour between high and low virulent VHSV. As you stated, the number of possible comparisons that can be done is very high and, even if the analysis of single genes could be very interesting, it is also out of scope of our work. Therefore, we leave this analysis for future works on this topic and, in any case, we provide all the relevant data in the supplementary tables (Table S5). Moreover, we integrated in the manuscript a new supplementary table (Table S6) with the top 10 up- and down-regulated genes for each timepoints related to both VHSV challenges (Lines 511-515).
Lines 365-444: Please remove references from this section. Authors must describe the results. The description of the genes and their functions should be included in the discussion section. As I comment before, in my opinion it is easy for the reader to know first which biological processes are modulated (activated/repressed) by the infection (a general idea) and then explain the genes involved in those processes.
The text was edited according to the Reviewer’s comment.
In my opinion this part of the manuscript should be revised in detail. All this information is already available in the table 1. It is surprising to me that the authors only focused on genes that appear in all sampling points and after filtering the data they only describe 39 genes for VHSV-H and 10 genes for VHSV-L. At this point I have no clear idea about which physiological processes are involved in the response to VHSV regardless of their pathogenicity.
Authors’ response. We thank the reviewer for the suggestion. There are several different ways to select the interesting genes to focus on. We chose to focus on the DE genes that appear in all sampling points because our main aim was to highlight the consistently different behaviour between high and low virulent VHSV. We are aware that genes appearing only at specific timepoints could be equally interesting. However, as previously stated, the number of possible ways to select genes is very high and it is simply not possible to consider multiple ways of select genes in the same work. In order to give an idea of the physiological processes involved in the response to VHSV, we chose to adopt gene ontology and the enrichment of terms regarding to the biological processes. GO term enrichment results, which describe the general behaviour of the infection and also the differences between high and low virulent VHSVs, are reported in the final paragraphs of the results (3.6 and 3.7) since they are used with the purpose of providing an overall idea of what is going on, a task that is nearly impossible to accomplish by analysing the DE genes individually.
Lines 451: It is not clear which DEGs were used in this analysis. Did you use all the DEGs obtained in each sampling point? or did you use the selected genes described in the previous section (39 genes for VHSV-H and 10 genes for VHSV-L). Reading this section I assume that you use all the DEGs obtained for H and L. In my opinion those analysis are difficult to combine with the previous description of selected genes since different set of genes were used.
Authors’ response. At line 451 of the original manuscript, the paragraph 3.6 "Gene ontology enrichment analysis" is dedicated to the general result on the enrichment of GO terms starts. The enrichment analysis is always done taking into account all the genes analysed, so all genes present in the rainbow trout annotation. From a mathematical point of view, the enrichment is computed as the Fisher’s exact test on the fraction of DE genes on the total belonging to a particular GO term respect the fraction of DE genes on the total not belonging to the particular GO term considered. So all genes, both DE and not DE, are used for the analysis and this fact is the reason why GO enrichment is a way to sum up the behaviour of forty thousand genes taken together into consideration simultaneously.
I suggest explaining the GO terms before the description of particular genes.
See response to the comment number 1 above in the “Review” section.
Lines 451-525: This section is too long. Authors should mix the numerical information with the results presented in the graph and delete all the duplicated information. The discussion of the results must be moved to the next section. In my opinion it is obvious that the increment of DEGs (figure 4) will also produce an increment of the GO terms in the enrichment analysis. I think that section 3.6 should be deeply reduced and combined with the next section (3.7)
Authors’ response. We thank the reviewer for the observation. Indeed, although some degree of proportionality is expected between number of DEGs and number of enriched GO terms, the two values are not in a linear relationship. From the genetic point of view, the number of associated GO to a single gene is highly variable and may vary from zero to as many as possible. Regarding ontology, a GO can be associated to a variable number of genes, again from zero to as many as possible (in this case obviously limited by the number of genes annotated in the analysed genome). Thus, the final result highly depends from which genes are differentially expressed and how they are distributed among GO terms, so it is something worthy to note if there is a common trend between number of DEGs and number of GO term enriched, as well as if a small number of DEGs results in an higher number of GO term enriched than the expected one. Moreover, since we get many enriched GO terms, which represent the final sum up of the behaviour of the previously found differentially expressed genes, we chose to provide enrichment results into two separate paragraphs, one dedicates to a general behaviour (3.6) and one dedicates to truly distinguish between shared and virus-specific enriched functions (3.7). So, there was no duplicated information between the two sections.
Lines 489-495: This part should be deleted or reduced.
Authors’ response. In our opinion, it is not possible to reduce this part, since it is already a much pruned description on how gene ontology hierarchy is, which is crucial for understanding the analysis and the results presented in the current paragraph (3.7). To give readers the highest possible aid in understanding the paragraph, in particular to those not expert on how gene ontology works, we prefer to maintain this part as it is.
This is the final part of the results. At this point authors described the infection process (mortalities and viral load), the biological processes that are activated after infection (enrichment analysis) and the genes that are present in all sampling points. I think that the information about specific genes is not well integrated in the text. Authors should analyze the set of genes with other comparison to have a better understanding of the infection process.
I wonder which biological processes are active during VHSV-H and L infection? Are they the same? Do they appear at the same time point or at different times? Could they be activated when the viral load reaches a minimum amount? Some of these questions are described in the text.
Authors’ response: We thank the reviewer for his/her opinion. The following answer is for both comments. As previously stated, there are multiple ways to select and analyse single genes and we chose the way that, in our view, fitted the best with our aims, which were to provide a general view about the progression of the infection and to highlight general differences between high and low virulent VHSVs. In any case, we provide all the results for all the genes in all the conditions tested for future works and for others groups with the hope that our data could be helpful for them.
Biological processes active during infection are described by enriched GO terms (paragraph 3.6 and 3.7). There are some enriched GO terms in common between VHSV-H and L infection, mostly related to immune system, and some are specific to one of the infection, in particular VHSV-H challenge show a higher number of specific enriched GO terms than VHSV-L challenge, probably indicating an higher immune stimulation linked to the high virulent virus respect to the low virulent one. Specific studies should be done in order to compute when a biological process activates concerning viral load. Our data indicate that at 1 day after infection, only VHSV-H induce the host answer, in total agreement with the fact that a 1 dpi only VHSV-H was detectable by the qRT-PCR we employed.
Which genes are involved in those processes? This information is not explored in the text since authors only analyze genes presented in the 3 sampling points. The description of genes should be improved by using additional comparisons. This information could be included in additional figures.
Authors’ response. We thank the reviewer for the final comments on results. About results on specific genes, as previously explained, we chose one of the many existing approach to analyze and present them; our decision was based on the fact that we think it fit the best on our purposes, which were mainly to find out general difference between low and high virulence VHSV. Different comparisons between set of genes will be left for future works which will focus on different aspect about VHSV infection. Biological processes are described by GO term enrichment analysis, so in paragraphs 3.6 and 3.7, which are made long with the clear aim of provide a comprehensive description of then, together with the provided tables (S7 and S8). As described in the material and methods section (2.9), GO term assignment to rainbow trout genes was downloaded from Ensembl (https://www.ensembl.org/index.html) using the Biomart utility integrated in the site. Such assignment was also merged with the one computed using Blast2GO, a tool which, starting for the blast results of a protein, infer homology to other genes which GO terms are known and retrieve them for the gene encoding the protein. Ensembl GO term assignment was already pretty complete, so Blast2GO integration only add very few genes, although we performed it in order to have the most complete GO term assignment, considering the limits of the process (e.g. only genes encoding protein can be considered). We are aware that it is not easy task to link all the information together, e.g link genes to gene ontology, so for this purpose we provided some supplementary tables (names S7 and S8) to aid in case a reader will be interested in particular genes of GO. However, as previously stated, we were interested in the general trend and so we provide the general results explaining the overall behaviour in case of VHSV infection, so intentionally we did not go deeply into particular results on specific genes. We hope that our work will be of greatly help for researcher wanting to study a particular gene of process in rainbow trout and/or VSHV infection.
In my opinion figures 3, 4 and 5 should be combined in a unique multi-panel figure.
Authors’ response: For a clear view of the images, we would definitively prefer to keep them separated.
Discussion:
Line 546: Too many references to describe the HK and spleen as the main hematopoietic organs. Please, review the references and select the most important.
Authors’ response: according to the reviewer’s request we kept only two references.
Lines 546-549: This result is quite interesting since it reflects that both organs are targets for the viral infection. It is frequently said that the HK is the main target organ, but your results also suggest that VHSV replicates in the SPL as efficient as in the HK.
Please see the answer to the comment above (on Results section for Line 291).
Lines 551-552: Differences in viral load between H and L could be related with different ability to replicate inside the fish. Why did the authors specify that the “absorption capacity” is altered? It could be possible that other steps in the viral cycle be affected by the genomic modifications described in those viral stains? Did the in vitro studies in reference 19 use the same viral strains described in the present manuscript?
Authors’ response: Yes, in the cited paper the same viral strains as this work were used. It this paper the most important difference observed between highly pathogenicity and low pathogenic VHSv was in the absorption capacity of the tested strains; while no significant differences were observed between the kinetics of progeny production and viral replication and average titres. In the cited paper (now reference 21), the experiment were performed in cell culture only, while the present work was performed also in vivo, so we cannot exclude that other steps in the viral cycle inside the fish could be affected and this is why we modified the sentence in the manuscript accordingly to the reviewer’s suggestion (Lines 625-628).
Line 554: “VHSV-L and VHSV-H strains most likely activates the host response in a different way”. This sentence is already supported by the data presented in the manuscript (figure 6). VHSV-H induce several GO processes that are not activated by VHSV-L.
Authors’ response: The sentence was removed according to the Reviewer’s comment.
Line 559: Please delete “cellular”
The text was edited according to the Reviewer’s comment.
Line 560: Why did you specify “systemic infection”? Did you check the presence of VHSV in almost all fish organs? I suggest delete the “systemic” term.
The text was edited according to the Reviewer’s comment.
Line 573: “VSHV-H starts to shape the host cell metabolism much earlier by requesting supplies for its own replication and to provoke the fish death”. Is this statement supported by the data? Which kind of metabolism is altered by the viral infection? Please, specify it.
Authors’ response: The Authors wish to thank the Reviewer for the comment. We agree that the sentence may confuse the readers; hence, it was deleted for clarity.
Line 575: I think that this is the weakness of the manuscript. Authors only analyze the DE genes that were expressed in all time points. Authors could analyze all the exclusive genes expressed at the different time points (in H and L infection) and see which GO terms are enriched in. Moreover they could compare the processes in H and L, which are specific for H and do not appear in L…Some of this information is already included in the manuscript.
Authors’ response: The Authors wish to thank the Reviewer for the comment. As previously stated, we chose to report the results only for genes that results differentially expressed at all time points since it was the most fitting way regarding our aims. To describe and analyse overall behaviour during VHSV infection, we employed gene ontology and enrichment on terms related to biological process, a type of analysis that takes into account all the genes annotated in a genome, both DE and not DE, so without any potential bias due to different ways to select genes and focus on them. As noted by the reviewer, some of the information that he/she mentioned are already included in the manuscript. In any case, we provide through various supplementary tables (namely S5 and Table S6) all the results related to all genes in all the comparisons done, so readers with aims different from ours could easily extract the information they need for their experiments.
Lines 557-580: is any previous study to support the participation of the TRIM63 gene in the anti-viral response?. I think that information in the lines 578-580 is not of great interest (heart) but TRIM63 and TRIM family proteins have been described to have emerging roles in innate immunity, since they are induced by IFNs (Ozato et al., 2008: doi: 10.1038/nri2413; Li et al., 2019: doi: 10.1016/j.prp.2019.152573).
Authors’ response: The Authors wish to thank the Reviewer for the comment. The text was implemented according to the Reviewer’s comment (lines 650-653).
Overall, the discussions section is well developed and it is easy to follow. I only want to highlight that almost all the description is speculative and is based on the bibliographic information using other viral and animal models. The discussion of the present manuscript could be stronger if authors include experimental data that support most of the suggestion.
Authors’ response: The Authors wish to thank the Reviewer for the observation. We also would like to highlight the novelty of the present study as we are reporting for the first time the transcriptomic response of rainbow trout infected with high and low virulent VHSV strains. Unfortunately, the scarcity of other transcriptomic studies based on VHSV, in rainbow trout or other fish species, led us to discuss our results basing on the bibliographic information. However, the herein presented data, which surely need to be confirmed and validated, are a starting point which will help future studies whose aim will be to deepen the comprehension and to better clarify the patterns of molecular responses induced by VHSV infection.

Round 2
Reviewer 2 Report
Comments and Suggestions for Authors
Authors have modified the manuscript according to the suggestions. This work should be published in the current format